# Application of regional meteorology and air quality models based on MIPS CPU Platform

**Zehua Bai[1,2], Qizhong Wu[1,2], Kai Cao[1,2], Yiming Sun[3], Huaqiong Cheng[1,2]**

[1]College of Global Change and Earth System Science, Faculty of Geographical Science, Beijing Normal University, Beijing 100875, China.

[2]Joint Center for Earth System Modeling and High Performance Computing, Beijing Normal University, Beijing 100875, China.

[3]Beijing Institute of Talent Development Strategy, Beijing 100032, China.

**Correspondence:** Qizhong Wu (wqizhong@bnu.edu.cn)

**Abstract.** The MIPS processor architecture is a type of Reduced Instruction Set Computing (RISC) processor architecture, which has advantages in terms of energy consumption and efficiency. There are few studies on the application of MIPS CPUs in the geoscientific numerical models. In this study, Loongson 3A4000 CPU platform with MIPS64 architecture was used to establish the runtime environment for the air quality modelling system WRF-CAMx in Beijing-Tianjin-Hebei region. The results show that the relative errors for the major species ($NO_2$, $SO_2$, $O_3$, CO, $PNO_3$ and $PSO_4$) between the MIPS and X86 benchmark platform are within ±0.1%. The maximum Mean Absolute Error (MAE) of major species ranged to $10^{-2}$ ppbV or μg $m^{-3}$, the maximum Root Mean Square Error (RMSE) ranged to $10^{-1}$ ppbV or μg $m^{-3}$, and the Mean Absolute Percentage Error (MAPE) remained within 0.5%. The CAMx takes about 15.2 minutes on Loongson 3A4000 CPU and 4.8 minutes on Intel Xeon E5-2697 v4 CPU, when simulating a 2h-case with four parallel processes using MPICH. As a result, the single-core computing capability of Loongson 3A4000 CPU for the WRF-CAMx modeling system is about one-third of Intel Xeon E5-2697 v4 CPU, but the thermal design power (TDP) of Loongson 3A4000 is 30W, only about one-fifth of Intel Xeon E5-2697 v4,

which TDP is 145W. Thus, Loongson 3A4000 has higher energy efficiency in the application of the WRF-CAMx modeling system. The results also verify the feasibility of cross-platform porting and the scientific usability of the ported model. This study provides a technical foundation for the porting and optimization of numerical models based on MIPS or other RISC platforms.

## 1 Introduction

In the recent years, with the increasing demand for high-performance computing resources and rapid development in the computer industry, especially supercomputer, central processing unit (CPU) has undergone significant advancements in logical structure, operational efficiency, and functional capabilities, making it the core component of current computer technology development. There are two main types: one is complex instruction set computer (CISC) CPU (George, 1990; Shi, 2008), mainly using X86 architecture, representative vendors including Intel, AMD, etc., and widely used in high-performance computing platforms. The other is reduced instruction set computer (RISC) CPU (Mallach, 1991; Liu et al.,2022), mainly using ARM, MIPS, RISC-V and other architectures, representative vendors including Loongson, etc., and mainly used in high-performance computing platforms, which have high efficiency, excellent stability and scalability. The Microprocessor without interlocked piped stages (MIPS) architecture is one of the significant representatives of RISC architecture. MIPS was originally developed in the early 1980s by Professor Hennessy at Stanford University and his group (Hennessy et al., 1982). The simplicity of the MIPS instruction set contributes to its ability to process instructions quickly, thus achieving higher performance even in low-power conditions. In 1999, MIPS Technology Inc. released the MIPS32 and MIPS64 architecture standard (MIPS Technology Inc., 2014). Compared to the CISC CPUs, RISC CPUs demonstrate excellent performance and power efficiency, which have gained popularity among chip manufacturers.

The Loongson processor family developed by Loongson Technology is mainly designed using MIPS architecture and Linux operating system (Hu et al, 2011), which

has rich application tools in Linux open-source projects. The main reason that currently restricts the development of CPUs that implement non-X86 instruction set architecture such as MIPS64 is the immature software ecosystem (Hu et al., 2016). Based on the strategy of open-source software, Loongson platform has gained abundant software tools, making it possible to further develop scientific computing and numerical models.

Air quality model (AQM) systems use mathematical equations and algorithms to simulate and predict the pollutant concentration in the atmosphere. The current AQMs have become more complex, incorporating numerous factors such as emissions from industrial sources, vehicle traffic, and natural sources, as well as meteorological conditions, including modeling meteorology, emissions, chemical reactions, and removal processes (Zhang et al., 2012). Regional-scale AQMs have been widely used to predict air quality in cities, formulate emission reduction strategies, and evaluate the effectiveness of control polices (Wang et al., 2023), including the Community Multiscale Air Quality (CMAQ) modelling system (Appel et al., 2017; Appel et al., 2021), the Comprehensive Air Quality Model with extensions (CAMx; RAMBOLL ENVIRON Inc., 2014), and the Nested Air Quality Prediction Modeling System (Wang et al., 2006; Chen et al., 2015). Due to the requirement of meteorological input, commonly used offline meteorological models such as WRF (Michalakes et al., 2001) are coupled offline with the regional AQMs to provide meteorological and chemical forecast as the WRF-AQM modeling system, such the WRF-CMAQ modeling system (Wu et al., 2014).

Both the meteorological and air quality numerical simulation rely heavily on high-performance computing systems. The WRF-AQM systems can run stably on high-performance computing platforms based on X86 or X86-compatible instruction set architecture (ISA) CPUs, which account for the highest percentage among the main processors of current high performance computing platforms. There are relatively limited researches on the application of WRF-AQM system on MIPS CPU platforms at present, this study focuses on the application of WRF-CAMx model on Loongson CPU platform based on the MIPS architecture. A simulation case covering the Beijing-Tianjin-Hebei region was set up to evaluate the differences and performance between

MIPS and X86 platforms. This study validated the stability of scientific computing on MIPS CPU platform, and it offered technical references and evaluation methods for the porting and application of numerical models on non-X86 platforms.

The remainder is organized as follows. Section 2 provides the model descriptions of the Weather Research and Forecasting–Comprehensive Air Quality Model with extensions (WRF-CAMx) modeling system, and the platform descriptions of both MIPS CPU platform and benchmark platform. The configuration of the air quality numerical simulation system and simulation case are also presented in Section 2. Section 3 describes porting and optimization of the WRF-CAMx modelling system on MIPS CPU platform. Section 4 analyzes the differences of model results between MIPS CPU platform and the benchmark platform. Section 5 discusses MIPS CPU performance in scientific computing. The conclusions are presented in Section 6.

## 2 Model and Porting Platform Description

The air quality modeling system was constructed using the WRF v4.0 model developed by National Center for Atmospheric Research (NCAR) (Skamarock et al., 2019), and the CAMx v6.10 developed by Ramboll Environment (RAMBOLL ENVIRON Inc., 2014), as shown in Figure 1. And the Loongson 3A4000 CPU platform was chosen for the porting work in the study. This study introduced the porting of WRF-CAMx modeling system to MIPS CPU platform.

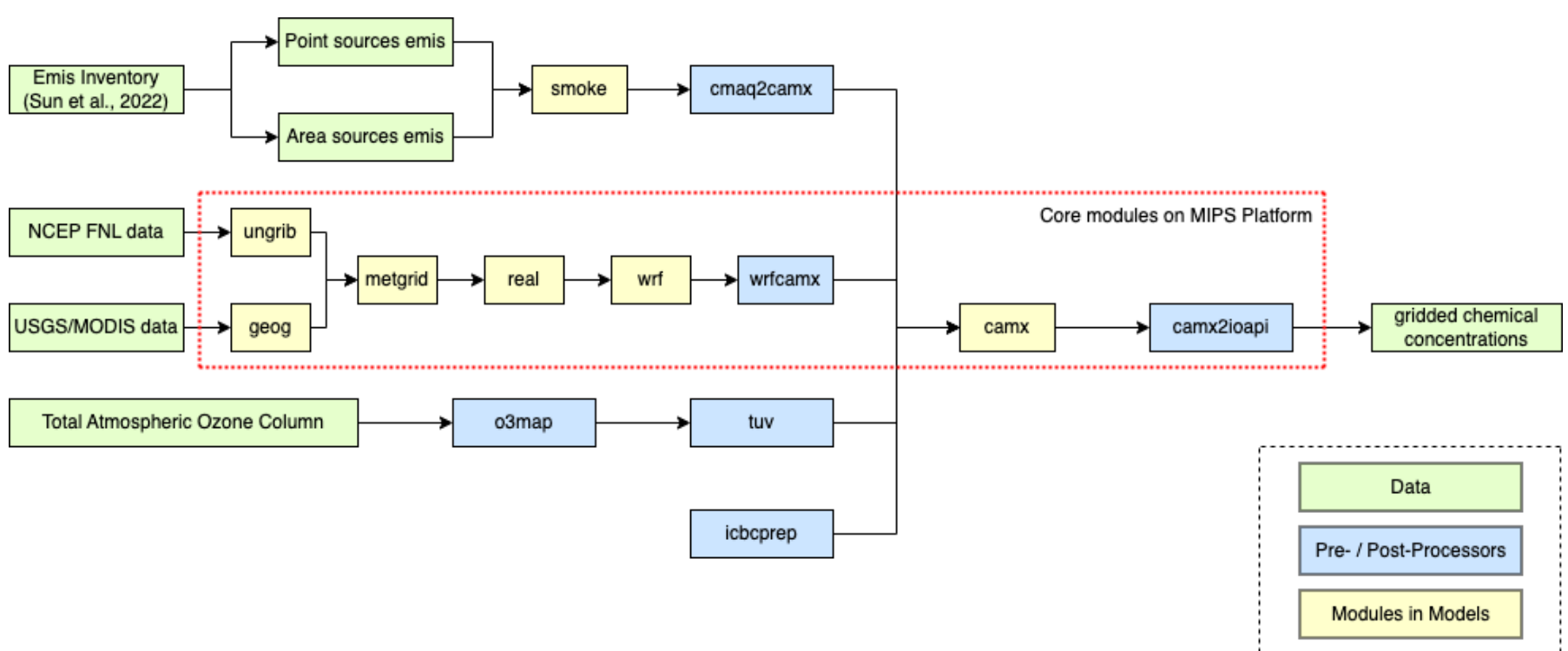

**Figure 1.** Framework of WRF-CAMx modeling system. The core modules have been ported to MIPS CPU platform. The core modules are framed by red dashed line in the

figure.

In Xi'an, China and Milan, Europe, the WRF-CAMx modelling system was developed, enabling high-resolution hourly model output of pollutant concentration within specific local urban areas (Pepe et al., 2016; Yang et al., 2020). The modeling system is widely used to study the spatial-temporal variation of pollutant concentration and source apportionment, analyze the contribution of regional transport to pollution and investigate the impact of initial conditions and emissions on pollution simulation in key regions such as the North China Plain, Sichuan Basin, and Fenwei Plain (Bai et al., 2021; Zhen et al., 2023; Zhang et al., 2022; Xiao et al., 2021).

**2.1 Description of WRF-CAMx modeling system**

WRF and CAMx serve as the core components of the modeling system. WRF is a high-resolution mesoscale model, which can be utilized for various purposes such as weather research and forecasting, physical parameterization scheme research, data assimilation and mesoscale climate simulation. In the modeling system, WRF provided gridded meteorological field data for air quality model CAMx. CAMx is an atmospheric pollutant calculation model, which can be utilized for simulating and predicting the concentrations of various air pollutants. The WRF and CAMx models are distinguished by modularity and parallelism, using MPI in parallel computing, making them efficient (Skamarock et al., 2019; RAMBOLL ENVIRON Inc., 2014).

In the modeling system, the SMOKE model and cmaq2camx program are used to process emission data and provide model-ready gridded emission data for the CAMx model. The wrfcamx program converts the WRF results into meteorological input files which are compatible with CAMx. TUV is a radiation transfer model capable of producing clean sky photolysis rate input files for the chemical mechanisms in CAMx, and the o3map program prepares ozone column input files for TUV and CAMx. The icbcprep program prepares initial and boundary condition files for CAMx with the profile, and the effects of initial conditions have been studied by Xiao et al. (2021). The camx2ioapi program converts the CAMx output files into netCDF format following the Models-3/IO-API convention, and then uses NCL or other softwares to analyses the

model results.

### 2.1.1 Model domain setup

The model domain focusing on the Beijing-Tianjin-Hebei region has been set up in this study. The WRF model has three nested domains with horizontal resolutions of 27km (D1), 9km (D2), and 3km (D3), as shown in Figure 2. The outer domain (D1) covers most parts of China, and the inner domain (D3) covers Beijing, Tianjin, and Hebei Province. The model domain is centered at (35°N, 110°E), with two true latitudes located at 20°N and 50°N. The vertical resolution of WRF is 34 vertical layers. The CAMx model has only one model domain, which is the innermost grid with a resolution of 3km (D3), mainly covering the Beijing-Tianjin-Hebei region. The vertical resolution of CAMx is 14 vertical layers, which is extracted from the WRF output files using the wrfcamx module, and the lower seven layers of CAMx are same as those in the WRF model.

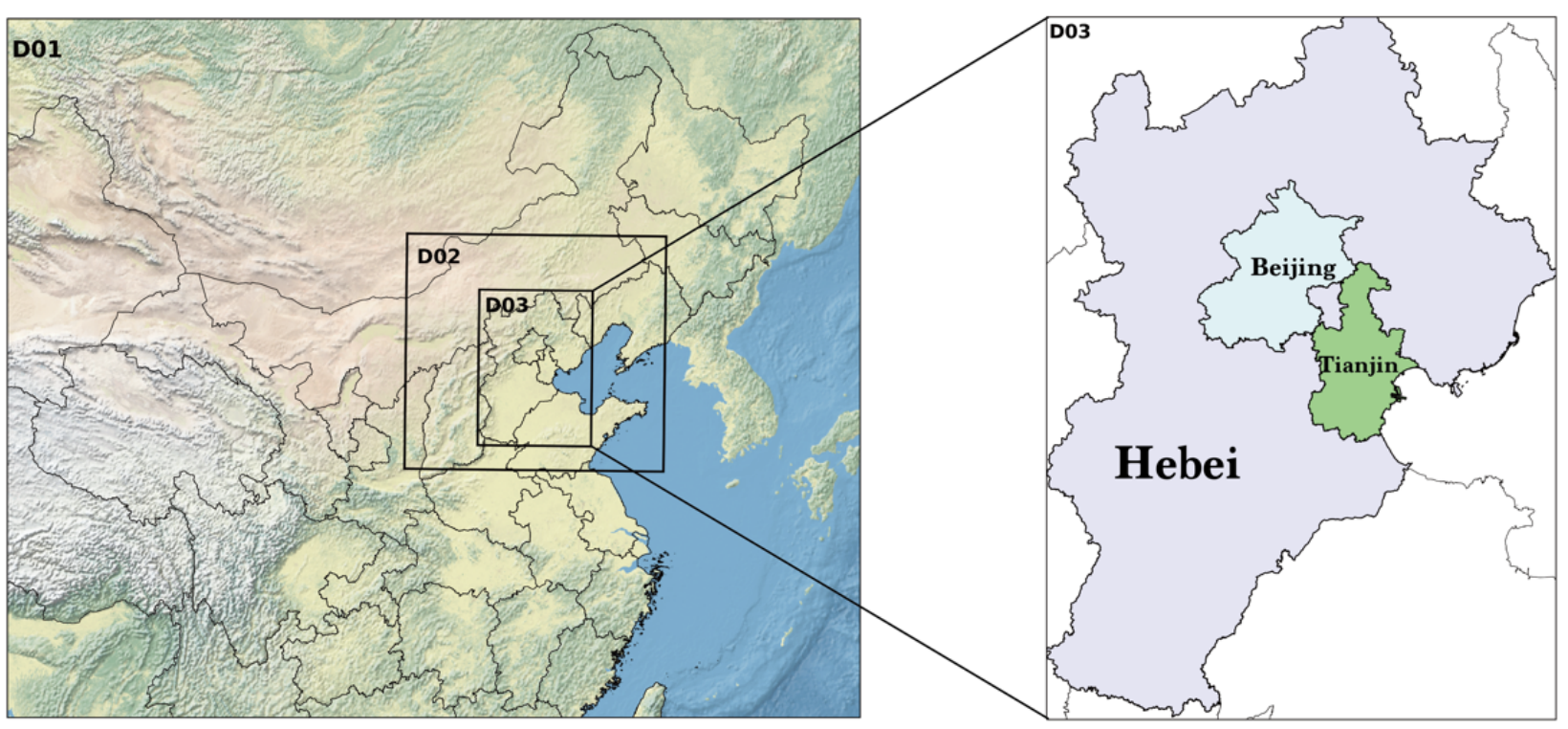

**Figure 2.** The domains of three-level nested grids in the WRF-CAMx modelling system. The respective horizontal resolutions are 27 km × 27 km (D1), 9 km × 9 km (D2), and 3 km × 3 km (D3).

### 2.1.2 Model configuration

For the meteorological model, the global meteorological initial and boundary fields for the WRF model are derived from the NCEP Global Final Reanalysis Data

(FNL), with a spatial resolution of 0.5° x 0.5° and a temporal resolution of 6 hours. And the parameterization schemes of the WRF model used in the simulation case are shown in Table 1.

For the air quality model, the meteorological files are provided by the WRF model are used for the chemical transport module in CAMx. The emission inventory used in the simulation case was obtained from Sun et al. (2022a). It contains basic emissions from Sun et al. (2022b) and fugitive dust emission from bare ground surfaces. The SMOKE model (v2.4) is used to process the emission inventory and provide gridded emissions for CAMx. The parameterization schemes of the CAMx model used in the simulation case are shown in Table 2.

**Table 1.** Parameterization schemes of WRF in research case.

| Parameterization process | Scheme |
|---|---|
| Microphysics | WSM3 |
| Longwave radiation | RRTM |
| Shortwave radiation | Dudhia |
| Land surface | Noah |
| Planetary boundary layer | YSU |
| Cumulus parameterization | Kain-Fritsch(new Eta) |

**Table 2.** Parameterization schemes of CAMx in research case.

| Parameterization process | Scheme |
|---|---|
| Horizontal Diffusion | PPM |
| Vertical Diffusion | K-theory |
| Dry Deposition | Zhang03 |
| Gas-phase chemical mechanism | CB05 |
| Aqueous aerosol chemistry | RADM-AQ |
| Inorganic gas-aerosol partitioning | ISORROPIA |

## 2.2 MIPS CPU platform description

Loongson CPU platform was chosen for the porting work in the study. Currently, the Loongson processor family has three generations of CPU products, evolving from single-core to multi-cores architectures and from experimental prototypes to mass-produced industrial products (Hu et al., 2011). The Loongson-2 processor is a 64-bit

general-purpose RISC processor series which is compatible with MIPS instruction set. It can be used in personal computers, mobile terminals, and various embedded applications, running many operating systems such as Linux and Android smoothly (Zhi et al., 2012). Wu et al. (2019) reports the application of the mesoscale model on Loongson 2F CPU platform. The Loongson-3 processor features a scalable multi-core architecture, targeting high-throughput data centers, high-performance scientific computing, and other applications, with the significant advantage of achieving a high peak performance-to-power ratio and striking a well-balanced trade-off between performance and power consumption (Hu et al., 2009).

A lot of porting and optimization research work has been conducted to ensure the proper functioning of the high-performance mathematical library on Loongson platforms, resulting in improved computing performance, such as FFT (Fast Fourier Transform) (Guo et al., 2012; Li et al., 2011; Zhao et al., 2012). The porting and optimization efforts conducted on the multi-core Loongson processors have successfully demonstrated the stability and efficiency in the numerical computing applications. These results provide valuable technical references and rationality validation for the numerical model application on Loongson platform.

The Loongson 3A series are multi-core processors designed for high-performance computers, featuring with high bandwidth, and low power consumption. The efficient design solution and the advantage of high energy efficiency ratio make servers based on Loongson CPUs highly competitive in performance, power consumption, and cost-effectiveness (Li et al., 2014; Wang et al., 2014). In this study, the Loongson platform uses the Debian Linux operating system, commercially known as Tongxin UOS (https://www.uniontech.com, last access: October 2023), and the Loongson 3A4000 processor, which is the first quad-core processor based on GS464v 64-bit microarchitecture in Loongson 3 Processor Family. The main technical parameters of Loongson 3A4000 CPU are shown in Table 3. Compared to previously released CPUs, the processor improves frequency and performance by optimizing on-chip interconnect and memory access path, integrating 64-bit DDR4 memory controller and on-chip security mechanism.

**Table 3.** Main Parameters of Loogson 3A4000 CPU*

| Loongson 3A4000 CPU Main Parameters | |
|---|---|
| **Main Frequency** | 1.8GHz–2.0GHz |
| **Peak Computing Speed** | 128GFlops@2.0GHz |
| **Transistor Technology** | 28nm |
| **Number of Cores** | 4 |
| **Processor Cores** | MIPS64 compatible<br>Support 128/256-bit vector instructions |
| **High-speed I/O** | 2 x 16-bit HyperTransport 3.0 control |
| **Typical Power Consumption** | <30W@1.5GHz<br><40W@1.8GHz<br><50W@2.0GHz |

*source: https://www.loongson.cn, last access: October 2023.

## 2.3 Benchmark platform description

This study uses an X86 CPU platform as benchmark platform compared to the MIPS CPU platform. The benchmark platform is powered by Intel Xeon E5-2697 v4 CPU, with strong floating-point performance and many technical features such as Intel Turbo Boost Technology (Intel Inc., 2023). The Intel Xeon E5-2697 v4 CPU has 18 cores, with 2.3GHz base frequency and 3.6GHz maximum Turbo Boost frequency, 45 MB Intel Smart Cache and 145W design power consumption. The operating system is CentOS Linux 7.4.1708. The main information for both platforms is shown in Table 4.

**Table 4.** The comparison of main configuration between MIPS and X86 platforms.

| | MIPS Platform | X86 platform |
|---|---|---|
| **CPU** | Loongson 3A4000 | Intel Xeon E5-2697 v4 |
| **Number of CPUs** | 1 | 1 |
| **Number of CPU cores** | 4 | 18 |
| **CPU Frequency** | 1.8GHz | 2.3GHz |
| **CPU instruction set** | MIPS64 | X86_64 |
| **Operating system** | Tongxin UOS | CentOS Linux 7.4.1708 |
| **Operating system kernel (Linux version)** | 4.19.0-loongson-3-desktop | 3.10.0-957.1.3.el7.x86_64 |

## 2.4 The difference between MIPS and X86 platforms

In this study, the numerical model's source code is written in Fortran, and commonly used compilers for X86 architecture include Intel Compiler, PGI and GNU Compiler. The compiler for MIPS platform is built using GCC 8.3 MIPS GNU/Linux cross-toolchain based on the open-source GNU Project, called MIPS GNU, and the latest version is 8.3. The compiler for the benchmark platform is set to X86 GNU, and the version is also 8.3. Table 5 shows the differences between the two platforms' GNU compilers in terms of applicable platforms. Compared to X86 GNU, the default compilation options of MIPS GNU compiler not only specify the platform architecture but also include additional instruction sets, such as atomic operation instruction set LLSC, shared library instruction set PLT, etc., which can optimize target programs compiled by GNU for MIPS architecture and improve computational efficiency.

**Table 5.** Comparison of GNU compiler between MIPS and X86 CPU platforms.

| Artitecture | MIPS64 | x86_64 |
|---|---|---|
| **Compiler** | MIPS GNU Fortran | X86 GNU Fortran |
| **Version** | 8.3 | 8.3 |
| **Target** | mips64el-linux-gnuabi64 | x86_64-redhat-linux |
| **Options (Architecture)** | -march=mips64r2 -mabi=64 | -march=x86-64 -mtune=generic |
| **Options (Instruction set)** | -mllsc -mplt -mmadd4 | / |
| **FLAGS(WRF)** | -fconvert=big-endian -frecord-marker=4 -ffree-line-length-none -O2 -ftree-vectorize -funroll-loops | |
| **FLAGS(CAMx)** | -fconvert=big-endian -frecord-marker=4 -ffixed-line-length-none -fno-align-commons -O2 | |

The WRF-CAMx modeling system depends on several scientific computing libraries. Firstly, the general data format libraries netCDF and HDF5 are required to store the large-scale gridded data for the modeling system. NetCDF is a self-describing data format developed by NCAR/Unidata, primarily used for storing multidimensional array data in fields like meteorology and earth sciences (UCAR/Unidata, 2021). HDF5 is a data format developed by HDF GROUP that supports complex data structures with multiple data types and multi-dimensional datasets (The HDF Group, 2019). In this study, netCDF-C (v4.8.1), netCDF-Fortran (v4.5.3), HDF5 (v1.12.1) and IOAPI (v3.1) were successfully installed on MIPS platform by building from their sources, which are

obtained from the official website.

The MPICH library is required to support parallel computing in the modeling system. In order to fully utilize computing resources, the method of MPI message communication is used in WRF and CAMx model (Wu et al., 2012). MPICH is an open-source, portable parallel computing library for implementing the MPI standard (Amer et al., 2021). It supports inter-process communication and data exchange in the parallel computing environment. Similarly, this study successfully installed MPICH (v3.4) on MIPS platform by building from its source. During the compilation and installation of the mentioned libraries above, the configure tool was used to check the basic information of the platform's CPU and compiler, and prepare for compatibility with platform before compilation, the GNU compiler is used to compile the source code of libraries, and the cmake tool is used to install the libraries. Additionally, the same runtime environment as MIPS platform was also built on the benchmark platform.

## 3 Porting the WRF-CAMx modelling system on MIPS CPU platform

The simulation result is influenced by several factors including processor architecture, operating system, compiler, parallel environment, and scientific computing libraries. In order to ensure stability and accuracy of numerical simulation, the models should be adapted to the new runtime environment when porting across platforms. Additionally, various operating systems have different tools, software and libraries, which may impact the results of numerical simulations.

In this study, the runtime environment for WRF-CAMx modeling system was built on MIPS platform. The configuration files for making the models were modified to fit the compiler of the UOS Linux system on MIPS platform. In order to verify the stability of scientific computing on MIPS platform, a control experiment was set up on the benchmark platform, minimizing the impact of other factors on simulation results of both platforms.

The WRF v4.0 and CAMx v6.10 were successfully deployed on MIPS platform through source code compilation and installation. In the WRF model, the default

options for GNU compiler which are suitable for MIPS architecture CPU are not provided in the configure file of the source code package, and it is necessary to manually add information about the CPU architecture, GNU compiler, and compilation flags on MIPS platform. Table 5 provides the detailed information added in the configure file, mainly about MIPS GNU Fortran. When compiling Fortran programs on MIPS platform, the MIPS GNU Fortran and necessary compilation flags must be specified. These flags include common Fortran file format flags such as -fconvert=big-endian and -frecord-marker=4, as well as optimization flags such as -O2 -ftree-vectorize -funroll-loops. By specifying the appropriate compiler and flags for MIPS architecture, the configure tool will provide necessary settings to compile WRF. Correspondingly, when compiling WRF on the benchmark platform, the compilation flags are strictly consistent with those of MIPS CPU platform, which ensures that differences in simulation results of two platforms are primarily attributed to the underlying hardware architecture rather than changes in compilation settings.

In the CAMx model, the makefile provides information about parallelism and compilers. Similarly, information about the CPU architecture, GNU compiler, and compilation flags on MIPS platform also needs to be added in the makefile. For the detailed information added in the makefile, please refer to Table 5. So far, the WRF-CAMx model has been successfully compiled and installed on the MIPS platform after modifications of the configuration files mentioned above.

## 4 The differences of model results on the two platforms

### 4.1 Validation of the spatial distribution

A simulation case has been designed to test the stability and availability of the WRF-CAMx modeling system on the MIPS CPU platform in Beijing. Starting from 00:00 on November 3, 2020, until 24:00 on November 5, 2020, the modelling system simulated the meteorological and air quality for a period of 72 hours, represents a moderate-sized real scientific workload, which allows for testing in a short time, and validating the results of the WRF-CAMx model on the MIPS platform and assessing

computational efficiency. By analyzing the differences in simulation results and computing time, the accuracy and performance of the modeling system on MIPS platform were evaluated, which further verifies the feasibility and stability of the modeling system after porting to the MIPS platform.

Common meteorological variables, including 2-meter temperature, land surface pressure, and relative humidity were selected to verify the WRF model results. Figure 3 shows the spatial distribution of the four meteorological variables after 72 hours simulation on different platforms, as well as the absolute errors (AEs). The meteorological variables from the modeling system on the different platforms exhibit a generally consistent spatial distribution in the Beijing-Tianjin-Hebei regions shown in Figure 3.

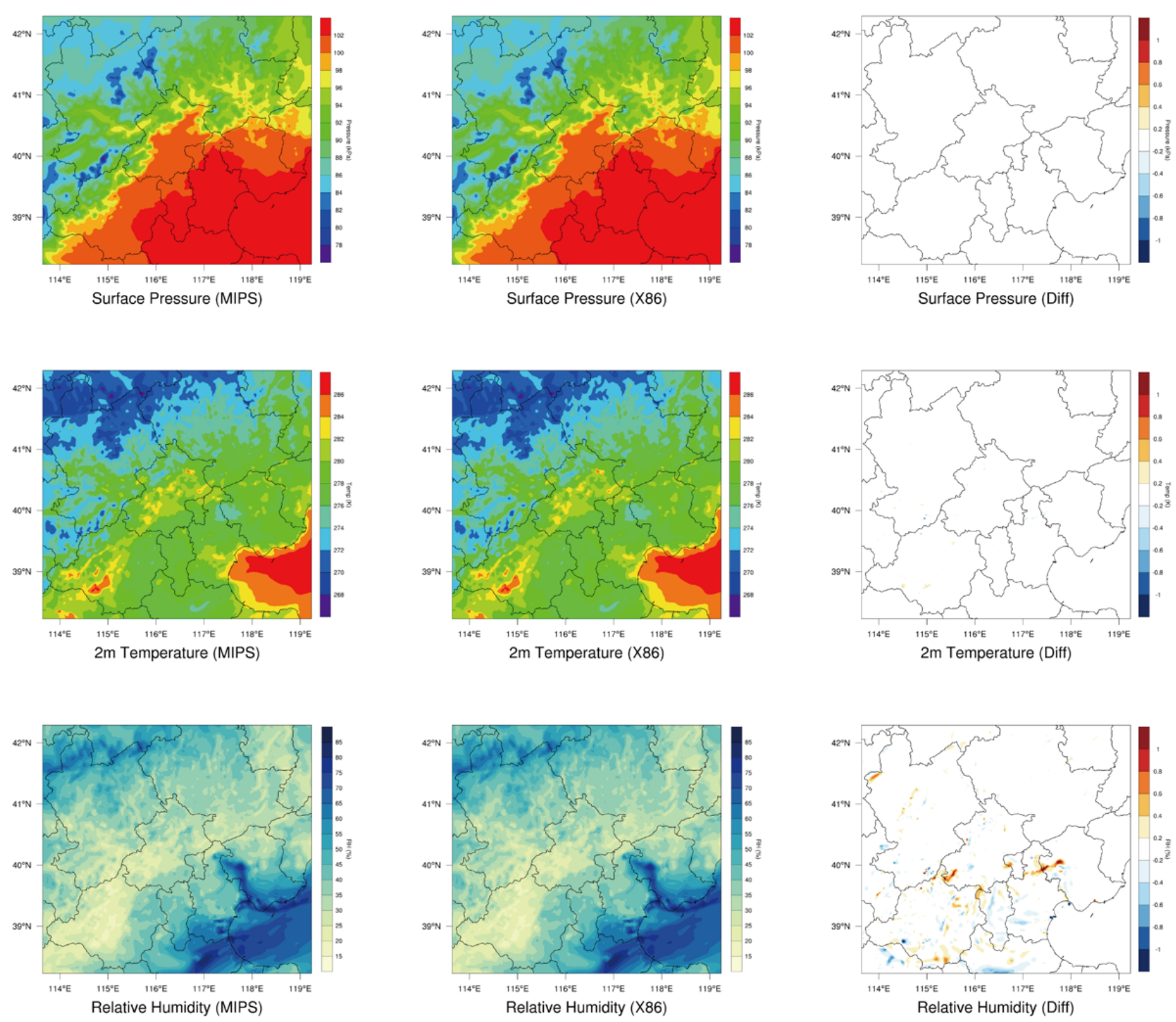

**Figure 3.** Spatial distribution of 2m temperature, surface pressure, relative humidity from WRF. Left column, MIPS platform. Middle, the X86 platform. Right, the differences between the MIPS and benchmark(X86) platform. Relative humidity is

calculated using the wrf-python package (Official website: https://wrf-python.readthedocs.io, last access: October 2023).

Similarly, the $NO_2$, $SO_2$, $O_3$, CO, $PNO_3$ and $PSO_4$ were selected to verify the CAMx model results on the MIPS platform. Figure 4 shows the spatial distribution of the six species, as well as the absolute errors (AEs) between the two platforms after 72 hours simulation. Simulating the 72h-case with four parallel processes using MPICH, CAMx takes about 9h on Loongson 3A4000 CPU and 2.6h on Intel Xeon E5-2697 v4 CPU. As shown in Figure 4, the spatial distribution of air pollution concentrations from the different platforms is essentially consistent, appearing very similar visually.

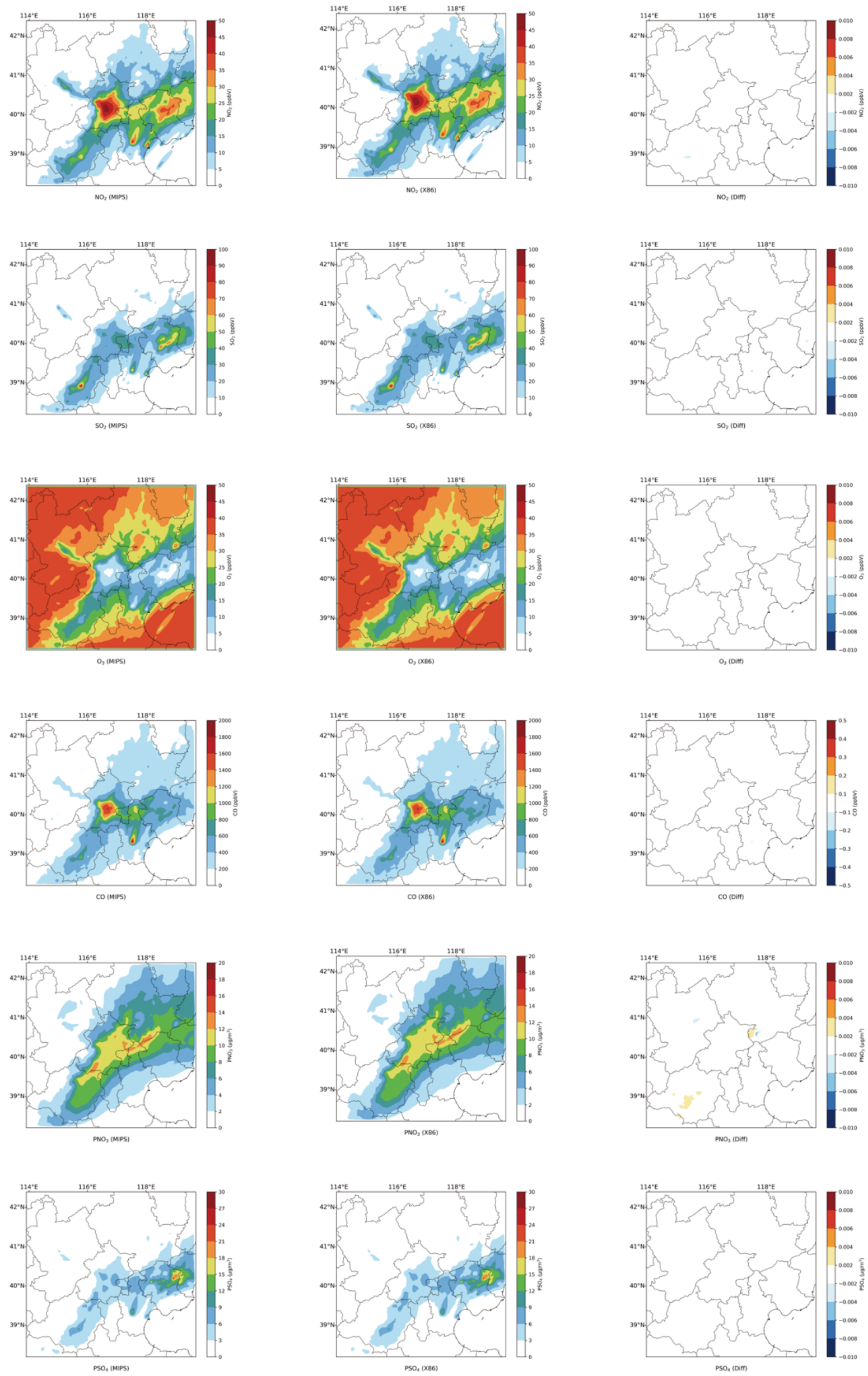

**Figure 4.** Spatial distribution of $NO_2$, $SO_2$, $O_3$, CO, $PNO_3$ and $PSO_4$ from CAMx on MIPS and benchmark platform. Left column, MIPS platform. Middle, the X86 platform. Right, the differences between the MIPS and benchmark(X86) platform.

As shown in Figure 5, the scatter plots between the two platform, it can be seen

that for the total of 22,765 grids within the 145x157 simulation domain, the root mean square errors (RMSEs) of the six species between the MIPS platform and benchmark platform are close to 0.001, which is essentially 0. The linear regression model was used to fit the scatters, and the regression slopes for each species are nearly 1, with intercepts close to 0, and the R2 values used for the goodness of fit are nearly 1. The fitted lines closely coincide with the "y=x" line, indicating that the differences between the MIPS and X86 platform for each species are minimal to negligible.

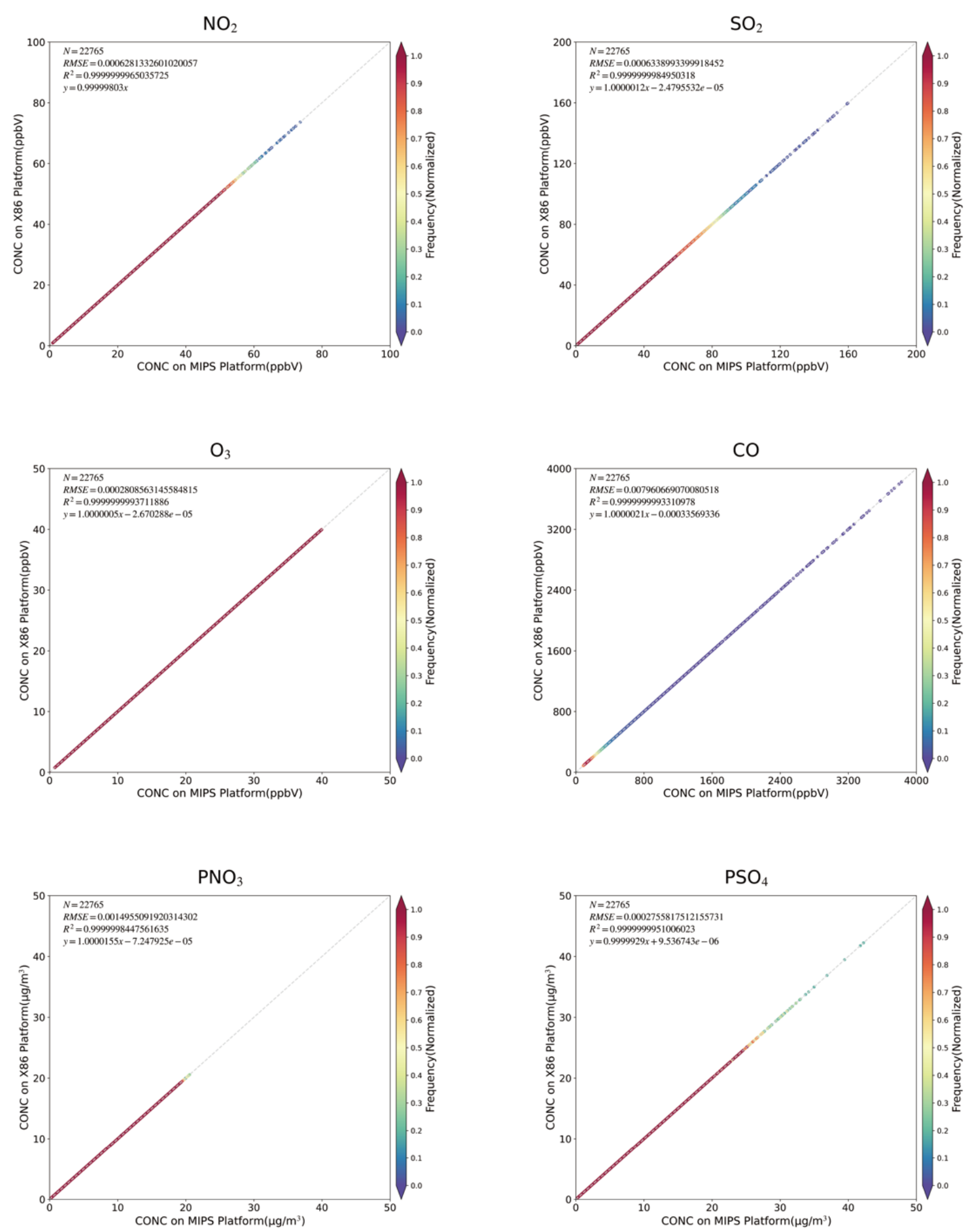

**Figure 5.** Scatter of grid concentrations for $NO_2$, $SO_2$, $O_3$, CO, $PNO_3$ and $PSO_4$ from CAMx on the MIPS and benchmark platform. The density of scatters is represented by

the colors.

Figure 6 is the boxplots which show the absolute errors (AE) and relative errors (RE) of the six species between MIPS and benchmark platform. According to Figure 6, the absolute errors of the six species are generally in the range of $\pm 10^{-3}$ ppbv (parts per billion by volume; the unit of $NO_2$, $SO_2$, $O_3$ and CO concentration) or μg $m^{-3}$(the unit of particle composition $PNO_3$ and $PSO_4$), and the relative errors are generally in the range of ±0.01%. Specially for CO, it exhibits more pronounced AEs compared to other species. In some grid boxes, the AEs between MIPS and benchmark platform exceed the range of $\pm 10^{-3}$ ppbv, but they remain in the range of $\pm 10^{-2}$ ppbv. In summary, there are some errors between the results of the modeling system on the MIPS and benchmark platform during the porting process. However, these errors are relatively minor compared to the numerical values. The reasons are attributed to the differences in the CPU architecture and compiler characteristics between the two platforms, such as data operations and precision running on different CPUs, which are primarily responsible for the observed errors.

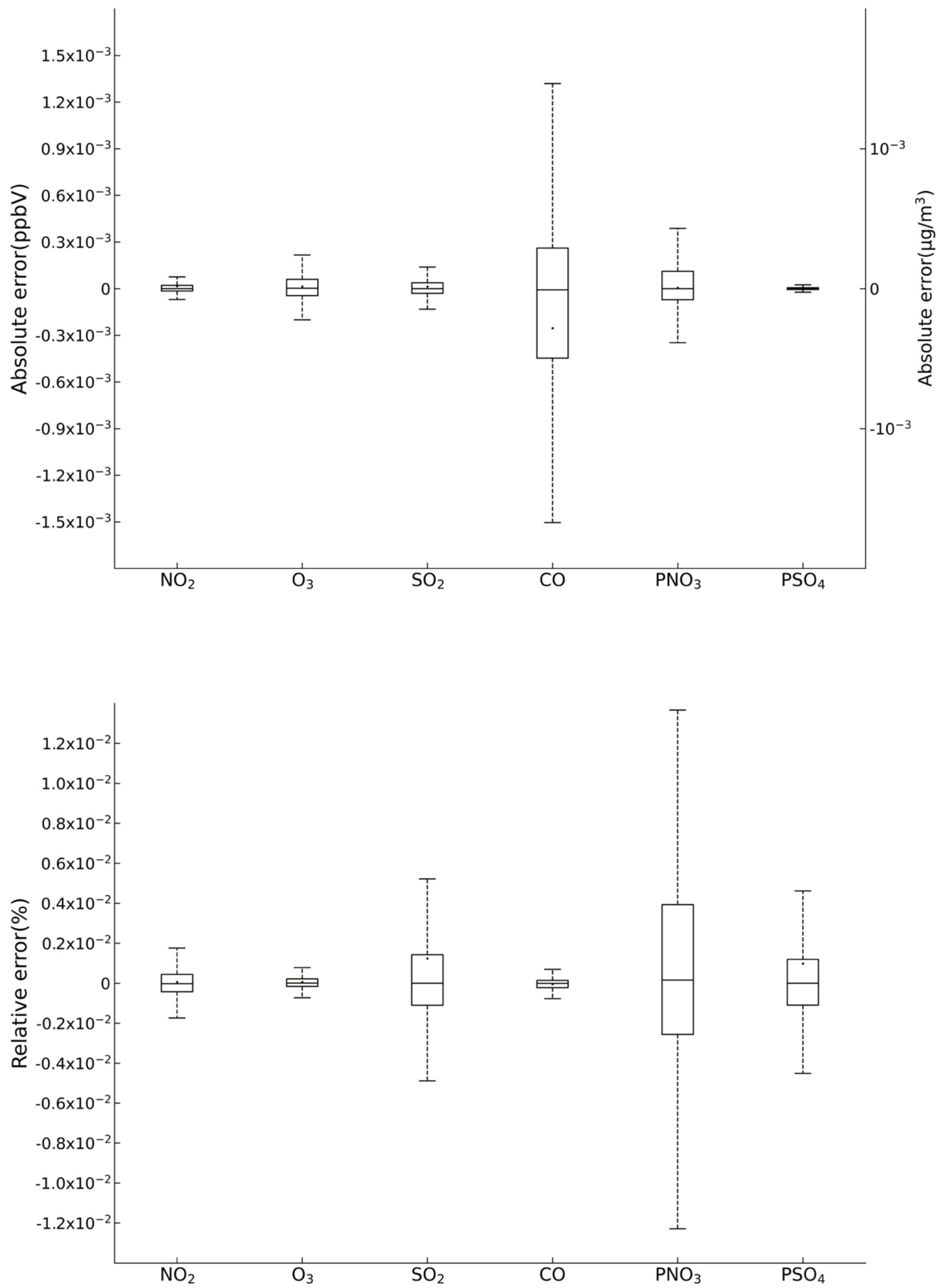

**Figure 6.** The absolute errors and relative errors for $NO_2$, $SO_2$, $O_3$, CO, $PNO_3$ and $PSO_4$ concentration in all grids between the MIPS and benchmark platform.

Additionally, random grids in the domain were selected to assess the precision of simulation results in localized regions. The positions of these grids were determined

based on 32 observation stations in Beijing, and the nearest grid was determined using the Euclidean Shortest Distance in the domain. The station map is presented in Figure S1 in the Supplement. The Taylor diagram is used to assess the precision of concentrations for six species near the observation stations, and the scatters representing the six species at 32 stations are highly overlapping. Statistical parameters used in the Taylor diagram, such as the correlation coefficient (R) approaching 1, normalized standard deviation (NSD) and normalized root mean square error (NRMSE) approaching 0, indicate high precision of the simulation results at specific stations on the MIPS platform.

### 4.2 Validation of the temporal distribution from the two platform

The time series of computational differences also be evaluated in this study. Random grid in the domain was selected to examine the hourly concentrations of the six species. Taking the example of the Beijing Olympic Center station (116.40 E, 39.99 N) from the National Standard Air Quality (NSAQ) stations, the time series of hourly concentrations in the grid of the Beijing Olympic Center station and relative errors between the MIPS and benchmark platform over the 72-hour period were shown in Figure 7. As shown in Figure 7, it can be seen that the time series of the air pollutant concentrations were highly consistent between the two platforms. In the 72-hour period, the relative errors for $NO_2$, $SO_2$, CO and $PSO_4$ remain in ±0.025%. For $PNO_3$, the relative errors remain in ±0.05%, and for $O_3$, they remain in ±0.1%. This indicates that the errors caused by different architectures are within a reasonable range.

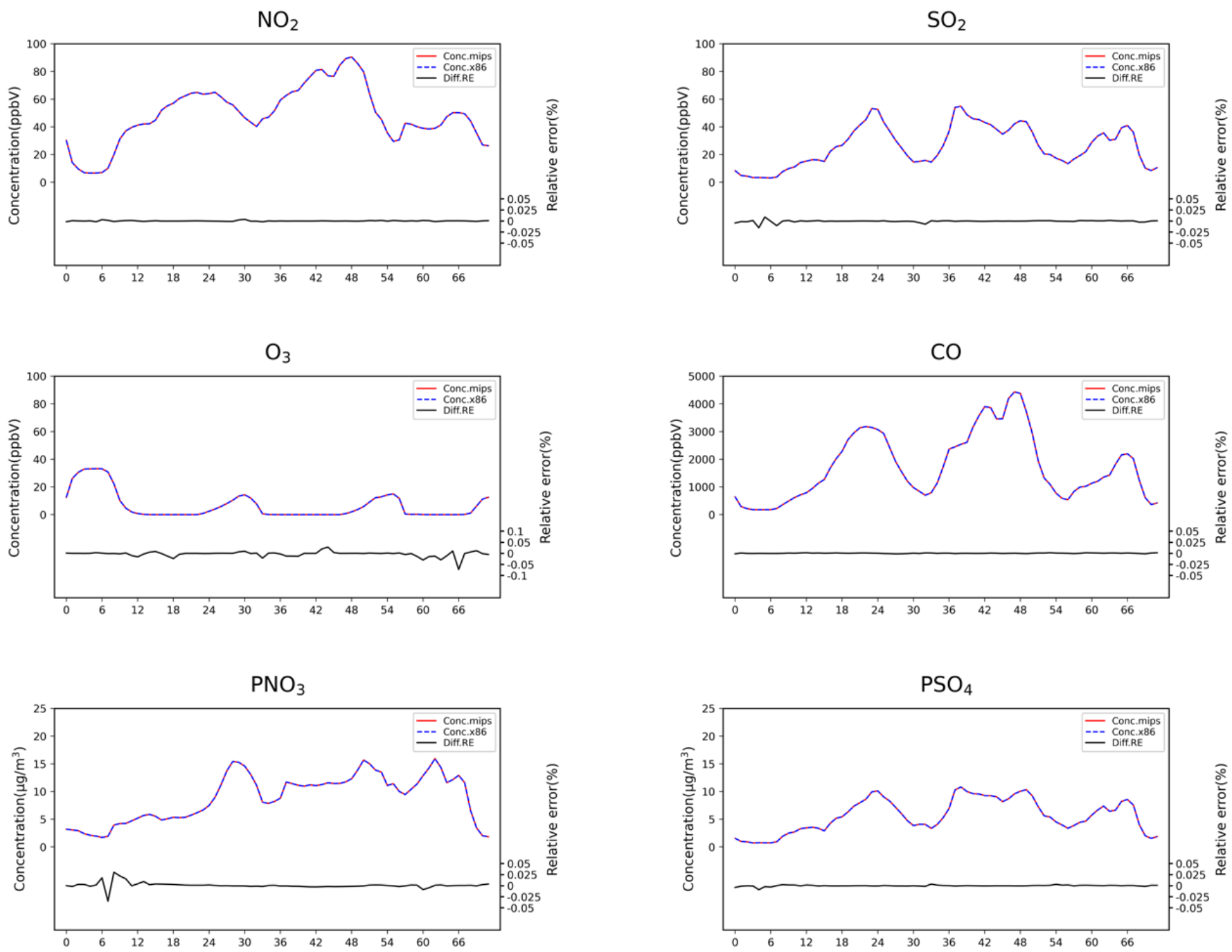

**Figure 7.** Time-series of $NO_2$, $SO_2$, $O_3$, CO, $PNO_3$ and $PSO_4$ concentrations and its relative errors (RE) at the Beijing Olympic Sports Center site between the MIPS and X86 platform. The red solid line and the blue dashed line, the CAMx model results on MIPS platform and X86 platform. The black solid line shows the relative errors (RE) between the MIPS and X86 platform.

To quantify the differences in the model results between the MIPS and benchmark platform, three statistical indicators are used to analyze the differences of concentration time series: Mean Absolute Error (MAE), Root Mean Square Error (RMSE), and Mean Absolute Percentage Error (MAPE). The MAPE quantifies the deviation between computational differences and simulated values. The smaller these indicators, the better accuracy and stability of scientific computing of the modeling system on the MIPS platform. The calculation formulas for these statistical indicators are provided in equations (1) to (3).

$$MAE = \frac{1}{n}\sum_{i=1}^{n}|MIPS(i) - Base(i)| \quad (1)$$

$$RMSE = \left[\frac{1}{n}\sum_{i=1}^{n}(MIPS(i) - Base(i))^2\right]^{\frac{1}{2}} \quad (2)$$

$$MAPE = \frac{1}{n}\sum_{i=1}^{n}\left|\frac{MIPS(i)-Base(i)}{MIPS(i)}\right| \times 100\% \quad (3)$$

In the equations, $n$ represents the number of grids in the domain. *MIPS(i)* represents the simulated value of a certain grid on the MIPS platform, and *Base(i)* represents the baseline value of a certain grid on the benchmark platform.

Figure 8 shows the time series of the concentration and their statistical indicators, MAE, RMSE, and MAPE during the 72-hour simulation. As show in the figure, for $NO_2$, $SO_2$, $O_3$, and $PSO_4$, the MAEs are all below $10^{-3}$ ppbv (μg $m^{-3}$), and the RMSEs are all below $10^{-3}$. The MAEs for CO and $PNO_3$ are below $10^{-2}$ ppbv (μg $m^{-3}$), and the RMSEs for $PNO_3$ are below $10^{-2}$, while the RMSEs for CO are below $10^{-1}$. This is because that $PNO_3$ and CO have relatively higher background concentrations compared to the other species. The MAPE of $PNO_3$ concentration mainly ranging in 0-0.5%, while the MAPE of CO concentration has the lowest values below 0.001%, and the other species are in the range of 0-0.01%. Overall, the above time-series analysis verifies the accuracy and stability of the modeling system on the MIPS platform.

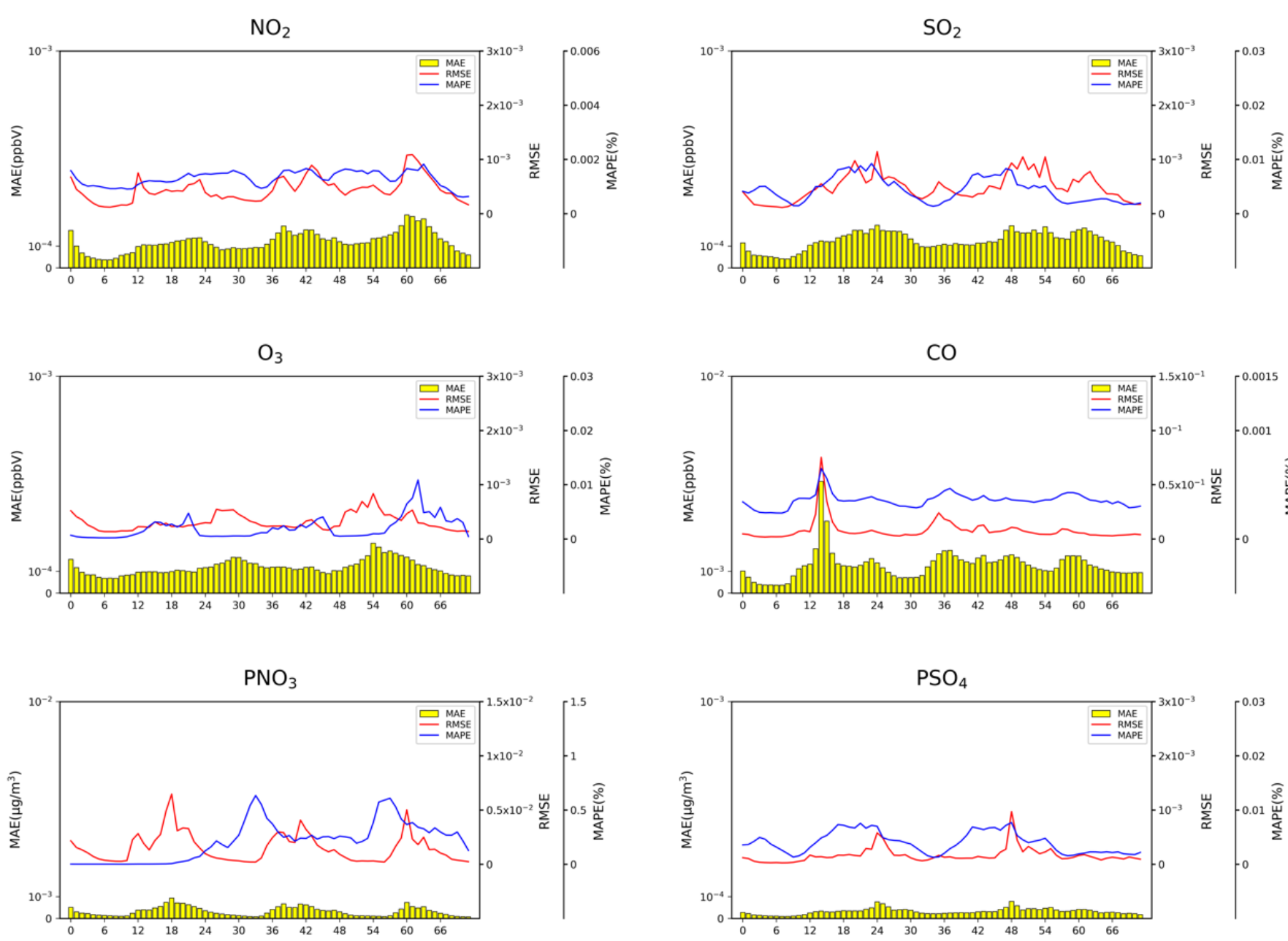

**Figure 8.** Time series of MAEs, RMSEs and MAPEs for $NO_2$, $SO_2$, $O_3$, CO, $PNO_3$ and

$PSO_4$ concentration in the 72h simulation. The yellow bar, the MAE. The red lines, RMSE, the blue lines, MAPE.

In this study, the evaluation method proposed by Wang et al. (2021) was also used to assess the scientific applicability of the model results on the MIPS platform. The RMSEs for $NO_2$, $SO_2$, $O_3$, CO, $PNO_3$ and $PSO_4$ concentration between the MIPS and benchmark platform were computed, along with the standard deviation used to describe the spatial variation of species, and the ratio of RMSE to std, as shown in Table 6. The differences of the four species between the two platforms are negligible compared to their own spatial variations. Therefore, the results on the MIPS platform meet the accuracy requirements for research purpose.

**Table 6.** RMSE, std, RMSE/std for $NO_2$, $SO_2$, $O_3$, CO, $PNO_3$ and $PSO_4$.

| | Differences in results<br>RMSE | Spatial variation<br>std | RMSE/std |
|---|---|---|---|
| **$NO_2$** | $6.3\times10^{-7}$ | 0.01 | $5.9\times10^{-5}$ |
| **$O_3$** | $2.8\times10^{-7}$ | 0.01 | $2.5\times10^{-5}$ |
| **$SO_2$** | $6.3\times10^{-7}$ | 0.02 | $3.9\times10^{-5}$ |
| **CO** | $7.9\times10^{-6}$ | 0.30 | $2.6\times10^{-5}$ |
| **$PNO_3$** | $1.5\times10^{-3}$ | 3.8 | $3.9\times10^{-4}$ |
| **$PSO_4$** | $2.7\times10^{-4}$ | 3.9 | $6.9\times10^{-5}$ |

In fact, the differences in model results cannot be completely eliminated, primarily due to the varying CPU architectures and compilers. In the practical applications, compared with the errors arising from the inherent uncertainties of the modeling system and the input data, the differences of model results between different platforms can even be considered negligible. The comprehensive analysis demonstrates that the results of the WRF-CAMx modeling system on the MIPS CPU platform are reasonable.

## 5 The evaluation about computational performance

Scientific computing involves a significant amount of floating-point operations, and the floating-point computational capability is a crucial indicator for CPU

performance. In this study, the simulation case was configured to conduct parallel computing tests on the MIPS and benchmark platform. These tests included assessing the CPU's single-core performance with the non-parallel model and the platform's parallel performance with the parallel model using multiple processes. The time of CAMx model running simulation case for 2 hours in the modeling system are shown in Figure 9. From the figure, it can be observed that under single-core conditions, the computing capability of the MIPS platform for CAMx is approximately one-third of the X86 benchmark platform.

It's worth noting that the simulation time of the CAMx model for running with two processes in parallel and running in non-parallel remains approximately consistent. This is because the MPI used in CAMx is designed using a "master/slave" parallel processing approach, and a process is allocated for input/output and message communication during the runtime (Cao K et al., 2023). This process doesn't perform any simulation in the model. Therefore, the time required for parallelism of two processes is comparable to the non-parallelism, and in some cases, it might even be slightly longer due to the overhead of MPI communication. Compared to non-parallel, the speedup of the MIPS platform with four-process parallelism using MPICH3 is approximately 2.8, while using OpenMP is about 2.9. For the X86 benchmark platform, running with four processes in parallel using MPICH3 has a speedup of approximately 2.7.

Additionally, the performance of the MIPS platform significantly decreases when the number of parallel processes exceeds 4. This is because the modeling system involves compute-intensive tasks. The Loongson 3A4000 CPU has four cores, and when the number of processes called by MPI matches the number of CPU cores, the CPU utilization can approach 100%. Further increasing the number of processes, the cores will compete for CPU resources, resulting in additional overhead and reduced computational efficiency.

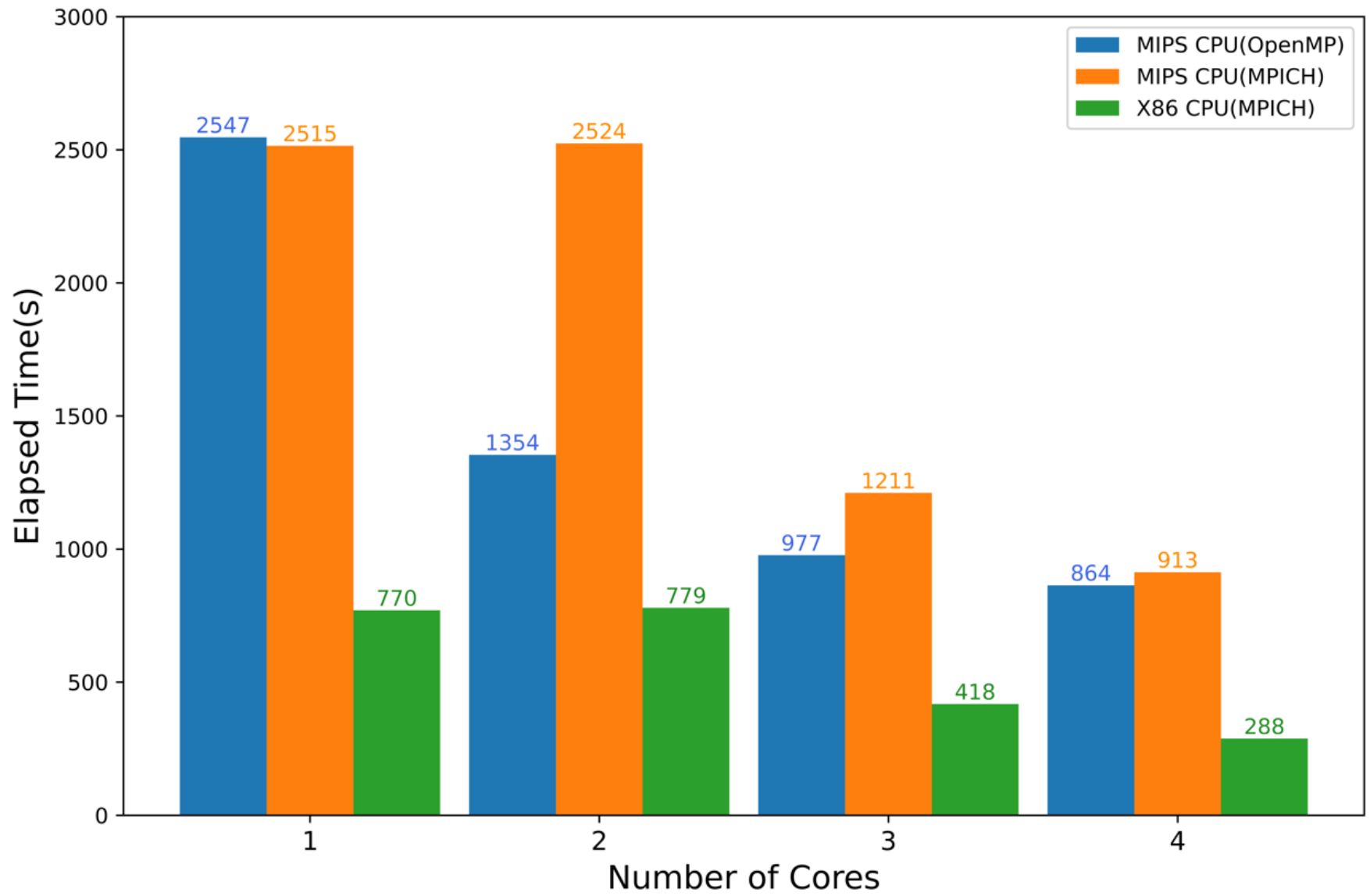

**Figure 9.** Elapsed time of CAMx model running simulation case for 2 hours on the MIPS and benchmark platforms.

In the recent years, the Longsoon CPUs have been continuously upgraded. Compared to the previous generations of products, the performance of the Longsoon 3A4000 CPU has shown significant improvement. Wu et al. (2019) simulated a nested domain covering Beijing for 48 hours using the MM5 model on the Longsoon 3A quad-core CPU platform. The results showed that the computational capacity of the Longsoon 3A platform for the MM5 model is approximately equivalent to around 1/12 of the Intel Core 2 Q8400 quad-core CPU, which was released in the same year. In the study of Luo et al. (2011), a comparison between Loongson 3A and Intel i5 was made by running NPB benchmark on each platform. The results shows that the performance of the 3A is nearly one-tenth of that of the i5. The rapid development of Loongson CPUs has provided a strong hardware foundation for the application of numerical simulation and scientific computing on MIPS architecture CPU platforms. The adaptation and optimization of the models based on MIPS CPUs will also be an important research direction in the future.

## 6 Conclusion

This study describes the application of the WRF-CAMx model on the MIPS CPU platform. The platform used in this study is Loongson 3A4000 quad-core 2.0GHz CPU, offering a peak operational speed of 128GFlops. It is equipped with the MIPS GNU compiler. The benchmark platform used the Intel Xeon E5-2697 v4 CPU along with the same version of X86 GNU compiler. Based on the characteristics of CPU architecture and compiler, this study has successfully completed the construction of runtime environment for the WRF-CAMx modeling system. The application of an air quality modelling system based on WRF-CAMx was successfully tested using a 72-hour simulation case in the Beijing-Tianjin-Hebei region.

The results showed that the spatial distribution of the meteorological variables and air pollutant species was nearly identical, with relative errors in the range of ±0.1%. Statistically, the maximum MAEs of major species ranged from $10^{-3}$ to $10^{-2}$ ppbv (μg $m^{-3}$), the maximum RMSEs ranged from $10^{-2}$ to $10^{-1}$ ppbv (μg $m^{-3}$), and the MAPEs remained within 0.5%, that the differences caused by the architectures and compilers were within a reasonable range. Simulating a 2h-case with four parallel processes using MPICH, CAMx takes about 15.2min on Loongson 3A4000 CPU and 4.8 min on Intel Xeon E5-2697 v4 CPU. In terms of single-core CPU performance, the single-core computing capability of Loongson 3A4000 CPU for the WRF-CAMx modeling system is about one-third of Intel Xeon E5-2697 v4 CPU.

Currently, Loongson Technology has introduced the LoongArch architecture which is compatible with MIPS, and it has been used in the next-generation product, the 3A5000 CPU (Hu et al., 2022). It is foreseeable that the LoongArch architecture will lead to more significant performance improvements. In the future, as the numerical models become more complex and computational scales become larger, more models will be tested on high-performance computing platforms equipped with the LoongArch architecture CPUs.

***Code and data availability.*** The source codes of CAMx version 6.10 are available at

https://camx-wp.azurewebsites.net/download/source (ENVIRON, 2023). The datasets related to this paper and the CAMx codes for MIPS CPU are available online via ZENODO (https://zenodo.org/records/10297970).

***Supplement.*** The supplement related to this article is available on-line.

***Author contributions.*** ZB and QW conducted the simulation and prepared the materials. QW planned and organized the project. ZB and QW completed the porting and application of the model for MIPS CPU. YS collected and prepared the emission data for the simulation. ZB, QW, KC, and HC participated in the discussion.

***Acknowledgements.*** The National Key R&D Program of China (2020YFA0607804) and the Beijing Advanced Innovation Program for Land Surface funded this work. The research is supported by the High Performance Scientific Computing Center (HSCC) of Beijing Normal University.

***Competing interests.*** The contact author has declared that none of the authors has any competing interests.

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
