# Peer review of "Application of regional meteorology and air quality models"

_EGUsphere, 2023_

## Author Comment (AC3)

**Reply to Referee 3:**

Thank you for the suggestions you provided. They are detailed and crucial for us to improve the manuscript.

**General comments**

Referee 3: The research paper focuses on the utilization of MIPS CPUs, particularly the Loongson 3A4000 CPU platform, in air quality prediction models. It evaluates the performance of the WRF-CAMx air quality modelling system in the Beijing-Tianjin-Hebei region using this platform. The study compares the MIPS CPU platform's performance with a benchmark X86 platform, analyzing various aspects like relative errors for major species, computational efficiency, and energy consumption. The results indicate the feasibility and efficiency of using MIPS architecture for such applications. This work has the potential to offer valuable guidance for using the MIPS platform for geoscientific modeling. I would suggest that the authors provide a more in-depth discussion on how to exploit the advantages of the MIPS platform. The structure of the paper could be improved, and additional tests are necessary to demonstrate the MIPS platform's performance.

**Reply:** Thank you very much for your appreciation of the potential practical value of this research. The MIPS platform with Loongson 3A4000 CPU in the manuscript has limitations in hardware performance and software ecosystem. Firstly, the platform only supports the GNU compiler and specific Linux operating system currently. Therefore, the performance evaluations can only be conducted in a certain environment. Secondly, due to the limited number of CPUs and cores on the platform used in this work, the maximum parallelism supported by the model system is 4. When the number of parallel processes exceeds 4, competition for computational resources restricts further performance evaluation of the platform. Additionally, the simulation case set up in this study fully utilize the computational resources of the platform, with the CPU utilization approaching 100% for each core during the simulation. Therefore, performance evaluation with higher parallelism and larger computational scales is challenging on this platform, but needs more computational resources, which exceeded our capability. However, the process-based MPI and the thread-based OpenMP were evaluated in the study. This provides a certain level of representativeness, as illustrated in "Figure 9" of the manuscript, as shown as Figure 1 in the followed. What's more, the lack of the effective software tools to provide more detailed assessments of the scientific computing capability on the platform is also an objectively existing issue. In the future, with the improvement of the ecosystem and performance of the MIPS and LoongArch platforms, additional tests will be considered to further demonstrate the platform's performance and discuss how to exploit the advantages. Recently, we acquired a platform equipped with the Loongson 3A6000 CPU, which is the latest product released by Loongson Technology. In order to enhance the universality of our research, we conducted the same tests based on this platform, the CAMx modeling system can run

stably on the LoongArch platform, and the performance evaluation is shown in Figure 2 in the followed. Relevant descriptions and results will be added to the revised manuscript. Additionally, the structure of the paper will be improved.

[Figure]

**Figure 1.** Elapsed time of CAMx model running simulation case for 2 hours on the MIPS and benchmark platforms.

[Figure]

**Figure 2**. Elapsed time of CAMx model running the same simulation case with MPICH

for 24 hours on the MIPS, LoongArch and benchmark platforms.

**Major comments and questions**

Referee 3: Every abbreviation that appears in the paper, including in the abstract and the main body, should be spelled out in full the first time it is used. For example, the abbreviations 'MIPS' and 'WRF-CAMx' are not spelled out in the abstract.

**Reply:** Thank you for the issues you pointed out. All abbreviations which appear for the first time will be carefully reviewed and corrected in the revised manuscript.

Referee 3: The model setups and analysis methods used in this paper should be presented prior to the results in Section 4. The content in Lines 305-309, 323-325, and 404-411 should be consolidated in Sections 2 or 3 as part of the methodology.

**Reply:** Thank you for the suggestions about the organization and structure of the article. These will be carefully considered and reflected in the improvement of the revised manuscript.

Referee 3: I am curious about the number of sockets available on the motherboard for the Loongson 3A4000 platform. Could the author conduct a larger-scale comparison using more Loongson 3A4000 CPUs compared to the X86 platform as shown in Figure 9?

**Reply:** The motherboard of the Loongson 3A4000 platform used in this study supports only one CPU with four physical cores. We will consider seeking platforms with more Loongson 3A4000 CPUs for larger-scale tests in the future.

Referee 3: Could the author investigate the impact of using different compilers or different compiler parameters on computational performance, in addition to the GNU?

**Reply:** Currently, the MIPS platform with Loongson 3A4000 CPU only supports the GNU compiler. Other compilers have not been adapted for the MIPS architecture. As for compiler parameters, the default options for MIPS GNU are fixed. They are used to specify the architecture of the target platform and optimize the target program based on specific instruction sets. There are no compiler parameters that significantly impact computational performance.

**Minor comments**

Referee 3: Line 92: Remove "The remainder is organized as follows."

**Reply:** This statement will be removed in the revised manuscript.

Referee 3: Line 113-115: Rephrase this sentence. The WRF is developed by NCAR, and CAMx is developed by Rambell. WRF-CAMx is just applied in Xi'an, China and Milan, Italy (not Europe).

**Reply:** This part of the statement is not clear and accurate enough. The intended meaning here is that the WRF-CAMx modeling system suitable for specific regions (such as Xi'an and Milan) has been applied in the research cited in the references. Relevant statements will be refined in the revised manuscript.

Referee 3: Line 123-126: The introduction for WRF is not professional. WRF is a meso-scale meteorology model, and you can use it for weather research and prediction. It can be used with a data assimilation technique, and testing its parameterization schemes is a way to improve WRF.

**Reply:** The description about the WRF model is not detailed and accurate enough, and it will be improved based on your suggestions in the revised manuscript.

Referee 3: Line 152-154: Why you used 14 layers not original 34 layers?

**Reply:** In practical applications of air quality simulation, we primarily focus on the concentration of air pollution species near the land surface. Therefore, the analysis of simulation results often involves extracting data from the near-surface layer for the CAMx model with wrfcamx module. This approach can significantly save computational resources in the air quality simulation.

Referee 3: Line 195-196: Was FFT used or related to this paper? If not, please remove it.

**Reply:** FFT was not used in this paper. The citation is intended to illustrate that there has been previous research on the application of scientific computing on MIPS platforms with Loongson CPUs. However, there is scarce research specifically focusing on large-scale applications like numerical models, with most studies centered around specific algorithms or programs (such as FFT). Therefore, we can only reference these studies. This part will be considered to remove in the revised manuscript.

Referee 3: Line 436: Give full names to RMSE, std. Why std are in lowercase but RMSE is not? Also, what's the statistic meaning or purpose of using the ratio of RMSE/STD?

**Reply:** The full names of RMSE (Root Mean Square Error) and std (standard deviation) will be added to the revised manuscript. The lowercase of std does not have a specific meaning. It is only used to distinguish it from statistical indicators such as MAE and RMSE, which are applied to analyze simulation differences between platforms. 'std' is only used to describe the dispersion of a certain species' concentration in the simulation

results. The ratio of RMSE/ STD does not have a specific statistical meaning. Wang et al. (2021) introduced it as a metric to assess the scientific usability of simulation results after model improvement. This has been recognized in academic research. In our study, the extremely small ratio of RMSE/STD means the simulation differences between the two platforms are negligible compared to the spatial variations of simulation results.

Citation: *Wang, P., Jiang, J., Lin, P., Ding, M., Wei, J., Zhang, F., Zhao, L., Li, Y., Yu, Z., Zheng, W., Yu, Y., Chi, X., and Liu, H.: The GPU version of LASG/IAP Climate System Ocean Model version 3 (LICOM3) under the heterogeneous-compute interface for portability (HIP) framework and its large-scale application, Geosci. Model Dev., 14, 2781–2799, https://doi.org/10.5194/gmd-14-2781-2021, 2021.*

---

## Author Response (AR3)

Q: There are a lot of Chinese remarks and revision notes in your revised manuscript, please revise and submit.

**Reply:** The remarks and revision notes in the track-changes manuscript has been translate into English, and resubmit. Thank you very much for your efforts.

Dear Editor,

Thank you for your efforts to improve our manuscript and the detailed and practical comments from three reviewers. We have made effort to revise our paper based on the reviewers' comments. The main changes in the revised paper are as follows:

1. Referee #1 mentioned the relationship between the LoongArch architecture and the MIPS architecture. In order to enhance the universality of our study, we have acquired a platform equipped with the latest released Loongson 3A6000CPU for the research of meteorology and air quality models on the LoongArch platform. Now the WRF-CAMx modeling system can also run stably on the LoongArch platform. Thus, the new results about the LoongArch architecture platform have been incorporated into the revised manuscript, and the title and main body have been updated accordingly. The description of the LoongArch platform has been added to Section 2, and the performance evaluation for the same simulation case has been added to Section 5 in the revised manuscript.

2. The structure of the article has been improved according to Referee #3' comment. The model configurations and analysis methods used in the manuscript have been uniformly described prior to the results analysis (Section 4) and computational performance evaluation (Section 5). For example, the description of statistical indicators used in Section 4 has been moved to Section 2 of the manuscript, and a subsection with the title "Statistical indicators for model results" has been created.

3. Some statements which are not clear or professional enough in the manuscript have been improved according to the referees' comments in the revised manuscript. For example, the abbreviations 'MIPS' and 'WRF-CAMx' have been spelled out in full the first time it is used in the abstract and main body of the revised manuscript. And the description of WRF and the citation for the WRF-CAMx modeling system has been improved.

We are thankful to the three referees for their thoughtful and constructive comments that help us improve the manuscript substantially, and the point-by-point responses to three referees' comments are given below.

**Response to Referee #1:**

Overall this article provides an interesting study to show case how a given system compares between different platforms. This work validate the robustness of the models.

**Reply:** Thank you for your comments on the manuscript. Your suggestions on certain

expressions in the manuscript are very insightful.

1. 3A4000 CPU works at 1.8-2.0GHz, it seems that the specific platform used for experiments are 1.8GHz. So when citing the power comsumption number, 40w instead of 30w should be used to be fair; and this statement in the Conclusion section also needs a correction: "The platform used in this study is Loongson 3A4000 quad-core 2.0GHz CPU, 497 offering a peak operational speed of 128GFlops"

**Reply:** Thank you for the constructive comment. The suggestions about the frequency, power consumption, and peak computing speed of the Loongson 3A4000 CPU are useful for us to improve the manuscript. Actually, based on the relevant information provided by the official website of Loongson Technology, the Loongson 3A4000 CPU can work within the frequency range of 1.5GHz ~ 2.0GHz, with 1.8GHz considered its base frequency. At this frequency, the thermal design power is 40W. Under extensive computational loads, its dynamic frequency can peak at 2.0GHz, with power consumption reaching 50W, achieving a peak computation speed of 128GFlops. In the comparison table of platform parameters (Table 4, lines 258-259 of revised manuscript), the base frequency of the 3A4000 CPU was selected to represent the platform's CPU frequency. Some expressions in the manuscript may not be sufficiently clear. We have already revised the relevant statements in lines 28-30 and 495-496 of original manuscript according to your suggestions, and new statements are in lines 32-34 and 559-560 of revised manuscript, which are as follows:

**lines 28-30 of original manuscript**: *but the thermal design power (TDP) of Loongson 3A4000 is 30W, only about one-fifth of Intel Xeon E5-2697 v4, which TDP is 145W.*

lines 32-34 of revised manuscript: *but the thermal design power (TDP) of Loongson 3A4000 is 40W, while the Loongson 3A6000 is 38W, only about one-fourth of Intel Xeon E5-2697 v4, whose TDP is 145W.*

**lines 495-496 of original manuscript**: *The platform used in this study is Loongson 3A4000 quad-core 2.0GHz CPU, offering a peak operational speed of 128GFlops.*

lines 559-560 of revised manuscript: *The platform used in this study is Loongson 3A4000 quad-core CPU with the main frequency of 1.8-2.0GHz, which can offer a peak operational speed of 128GFlops.*

2. The LoongArch architecture is not direct compatible with MIPS architecture. But Loongson does provide a binary translation software to run MIPS software with small performance loss.

**Reply:** Thank you for the information you provided about Loongson binary translation technology. LoongArch instruction set compatibility is achieved through Loongson's binary translation technology, allowing it to support instruction sets such as MIPS, X86,

ARM and others. In the LoongArch-based Linux operating system, Loongnix, provided by Loongson, not only can it execute native LoongArch programs, but it can also run software designed for Windows, Android, and Linux programs using MIPS, X86, ARM instruction sets through translation with slight efficiency loss. In order to better understand the application performance of the LoongArch architecture on the meteorology and air quality models, a platform equipped with the Loongson 3A6000 CPU was obtained recently, which is the latest product released by Loongson. To enhance the universality of our study, the WRF-CAMx model system was built on the LoongArch architecture platform using source code compilation and installation, instead of directly run MIPS modeling system using binary translation software. The CAMx modeling system can also run stably on the LoongArch platform, and the performance evaluation for the same simulation case are shown in Figure 1 and Figure 2 in the following.

[Figure]

Figure 1. Elapsed time of CAMx model running the same simulation case with MPICH for 24 hours on the MIPS, LoongArch and benchmark platforms.

[Figure]

Figure 2. Elapsed time of CAMx model running the same simulation case with OpenMP for 24 hours on the MIPS, LoongArch and benchmark platforms.

The time of CAMx model running simulation case for 24 hours in the modeling system are shown in Figure 1, it can be observed that the computing capability of the Loongson 3A6000 LoongArch platform for CAMx model is slightly lower than the E5-2697v4 (X86) benchmark platform under single-core conditions. Additionally, the Loongson 3A6000 CPU has four physical cores and eight logical cores, and when the number of processes called by MPI matches the number of physical cores, the computational load is evenly distributed across each core. Although the Loongson 3A6000 supports hyper-threading, further increasing the number of processes, CPU starts to schedule logical cores to allocate computational load. Thread scheduling will result in additional overhead and reduced computational efficiency.

The new results about LoongArch CPU platform will be incorporated into the revised manuscript. Thanks again for the expert's encouragement of this work. **The title of the original manuscript has been revised.** The new title is in lines 1-2 of revised manuscript, "and LoongArch" has been added into the title, which are as follows:

**Title, lines 1-2 of original manuscript:** *Application of regional meteorology and air quality models based on MIPS CPU Platform*

Title, lines 1-2 of revised manuscript: *Application of regional meteorology and air quality models based on MIPS and LoongArch CPU Platforms*

The main body of the manuscript has been revised, mainly includes:

(1) The performance of Loongson platforms has been updated in abstract (lines 24-26 of original manuscript and lines 27 -29 of revised manuscript):

[revised manuscript text omitted]

In lines 88, 90, 93, 98, 101, 103, 112, 115, 210, 250, 262, 290, 298 of revised manuscript, "MIPS" was replaced by "MIPS and LoongArch".

(3) The performance evaluation of LoongArch platform has been added into Section 5, and Figure 9 has been updated accordingly, which are as follows:

**lines 451-455 of original manuscript:** The time of CAMx model running simulation case for 2 hours in the modeling system are shown in Figure 9. From the figure, it can be observed that under single-core conditions, the computing capability of the MIPS platform for CAMx is approximately one-third of the X86 benchmark platform.

lines 498-502 of revised manuscript: *The time of CAMx model running simulation case for 24 hours in the modeling system are shown in Figure 10. From the figure, it can be observed that under single-core conditions, the computing capability of the MIPS platform for CAMx is approximately one-third of the X86 benchmark platform, and the LoongArch platform is slightly lower than the X86 benchmark platform.*

**lines 463-465 of original manuscript:** Compared to non-parallel, the speedup of the MIPS platform with four-process parallelism using MPICH3 is approximately 2.8, while using OpenMP is about 2.9.

lines 510-514 of revised manuscript: *Compared to non-parallel, the speedup of the MIPS platform with four-process parallelism using MPICH3 is approximately 2.8, while using OpenMP is about 2.9, and the speedup of the LoongArch platform with four-process parallelism using MPICH3 is approximately 2.8, while using OpenMP is about 2.9.*

**lines 472-474 of original manuscript:** Further increasing the number of processes, the cores will compete for CPU resources, resulting in additional overhead and reduced computational efficiency.

lines 520-531 of revised manuscript: *Further increasing the number of processes, the cores will compete for CPU resources, resulting in additional overhead and reduced computational efficiency. As for LoongArch platform, the performance slightly decreases when the number of parallel processes exceeds 4. The Loongson 3A6000 CPU has four physical cores and eight logical cores, and when the number of processes called by MPI matches the number of physical cores, the computational load is evenly distributed across each core. Although the Loongson 3A6000 supports hyper-threading, further increasing the number of processes, CPU starts to schedule logical cores to allocate computational load. Thread scheduling will result in additional overhead and reduced computational efficiency. This explains why the elapsed time is slightly higher when CAMx running with 5 parallel processes compared to 4 parallel processes as shown in the section 2 of Supplementary Material.*

**lines 476-477 of original manuscript:**

[Figure]

*Figure 9. Elapsed time of CAMx model running simulation case for 2 hours on the MIPS and benchmark platforms.*

lines 534-535 of revised manuscript:

[Figure]

[Figure]

**Figure 10.** *Elapsed time of CAMx model running simulation case with MPICH and OpenMP for 24 hours on the MIPS, LoongArch and benchmark platforms.*

**Response to Referee #2**

Thank you for your comments on the manuscript. The questions and suggestions you raised are very detailed and provide practical guidance.

Referee 2: This manuscript summarizes an exercise in porting the WRF-CAMx modeling system to a specific computational platform, namely Loongson 3A4000 CPU platform with MIPS64 architecture. Model simulations of evolution of air pollution over a period of 72-hour duration over a domain encompassing the Beijing-Tianjin-Hebei region are conducted on this platform and benchmarked against comparable simulations on a X86 platform. Additional simulations for a much shorter 2h period with CAMx are also conducted to examine parallel performance on the different architectures. The authors present a variety of standard statistical measures to demonstrate the cross-platform porting of the WRF-CAMx to the Longsoon 3A4000 platform they use. The manuscript is generally well written and clearly describes the work conducted by the authors. However, in my assessment the manuscript lacks scientific novelty and does little to advance either the development and evaluation of the modeling systems examined or in providing a robust assessment of the execution of these models on emerging architectures. In my view, much of what is presented, is a standard exercise in porting a numerical modeling system to a different computational platform and steps that are routinely undertaken to establish benchmarking on such systems – these tests are commonplace for both the WRF and the CAMx models used

here as well as other air pollution modeling systems and are routinely conducted by their respective user communities. While the successful porting of these specific models (WRF v4.0 and CAMxv6.10) to a specific platform (Loongson 3A4000 CPU) may likely be of interest to a segment of the users of these models who may also be planning to use these specific MIPS CPU platforms, such assessments are commonly discussed in the user forums of these (and similar) modeling systems – I thus struggle to identify the scientific and technical uniqueness of this contribution.

**Reply:** The main innovation of this article is the implementation of a relatively comprehensive, moderately scaled, and highly practical scientific computing application based on the MIPS platform, also a new architecture, named LoongArch and mentioned by another referee. In our study, we found that the same simulation case on the latest LoongArch platform, the elapsed time is nearly one third of that on the MIPS platform, while the power consumption is similar, which the TDP of Loongson 3A4000 is 40W and Loongson 3A6000 is 38W. Therefore, we believe that RISC architectures, such as MIPS and LoongArch, have great potential for scientific application in high-performance computing. Thus, the results about simulation on LoongArch platform have been added in the revised manuscript. The title, abstract, Section 2 for platform description and Section 5 for performance evaluation have been updated in the revised manuscript.

**Title, lines 1-2 of original manuscript:** *Application of regional meteorology and air quality models based on MIPS CPU Platform*

Title, lines 1-2 of revised manuscript: *Application of regional meteorology and air quality models based on MIPS and LoongArch CPU Platforms*

**Abstract, lines 24-26 of original manuscript:** *The CAMx takes about 15.2 minutes on Loongson 3A4000 CPU and 4.8 minutes on Intel Xeon E5-2697 v4 CPU, when simulating a 2h-case with four parallel processes using MPI.*

Abstract, lines 27-29 of revised manuscript: *The CAMx takes about 195 minutes on Loongson 3A4000 CPU, 71 minutes on Loongson 3A6000 CPU and 66 minutes on Intel Xeon E5-2697 v4 CPU, when simulating a 24h-case with four parallel processes using MPICH.*

The description of LoongArch platform has been added into Section 2, which are as follows. Table 3, Table 4 and Table 5 have been updated accordingly.

**Lines 210-213 of original manuscript:** Compared to previously released CPUs, the processor improves frequency and performance by optimizing on-chip interconnect and memory access path, integrating 64-bit DDR4 memory controller and on-chip security mechanism.

Lines 233-243 of revised manuscript: *Compared to previously released CPUs, the*

[revised manuscript text omitted]

In lines 88, 90, 93, 98, 101, 103, 112, 115, 210, 250, 262, 290, 298 of revised manuscript, "MIPS" was replaced by "MIPS and LoongArch".

The performance evaluation of LoongArch platform has been added into Section 5, and Figure 9 has been updated accordingly, which are as follows:

**lines 451-455 of original manuscript:** The time of CAMx model running simulation case for 2 hours in the modeling system are shown in Figure 9. From the figure, it can be observed that under single-core conditions, the computing capability of the MIPS platform for CAMx is approximately one-third of the X86 benchmark platform.

Lines 498-502 of revised manuscript: *The time of CAMx model running simulation case for 24 hours in the modeling system are shown in Figure 10. From the figure, it can be observed that under single-core conditions, the computing capability of the MIPS platform for CAMx is approximately one-third of the X86 benchmark platform, and the LoongArch platform is slightly lower than the X86 benchmark platform.*

**Lines 463-465 of original manuscript:** Compared to non-parallel, the speedup of the MIPS platform with four-process parallelism using MPICH3 is approximately 2.8, while using OpenMP is about 2.9.

lines 510-514 of revised manuscript: *Compared to non-parallel, the speedup of the MIPS platform with four-process parallelism using MPICH3 is approximately 2.8, while using OpenMP is about 2.9, and the speedup of the LoongArch platform with four-process parallelism using MPICH3 is approximately 2.8, while using OpenMP is about 2.9.*

**lines 472-474 of original manuscript:** Further increasing the number of processes, the cores will compete for CPU resources, resulting in additional overhead and reduced computational efficiency.

Lines 520-531 of revised manuscript: *Further increasing the number of processes, the cores will compete for CPU resources, resulting in additional overhead and reduced computational efficiency. As for LoongArch platform, the performance slightly decreases when the number of parallel processes exceeds 4. The Loongson 3A6000 CPU has four physical cores and eight logical cores, and when the number of processes called by MPI matches the number of physical cores, the computational load is evenly distributed across each core. Although the Loongson 3A6000 supports hyper-threading, further increasing the number of processes, CPU starts to schedule logical cores to allocate computational load. Thread scheduling will result in additional overhead and reduced computational efficiency. This explains why the elapsed time is slightly higher when CAMx running with 5 parallel processes compared to 4 parallel processes as*

*shown in the section 2 of Supplementary Material.*

**Lines 476-477 of original manuscript:**

[Figure]

*Figure 9. Elapsed time of CAMx model running simulation case for 2 hours on the MIPS and benchmark platforms.*

lines 534-535 of revised manuscript:

[Figure]

[Figure]

**Figure 10.** *Elapsed time of CAMx model running simulation case with MPICH and OpenMP for 24 hours on the MIPS, LoongArch and benchmark platforms.*

This work enhances the adaptability of the modeling system to emerging platforms, making the models more robust. RISC, as an emerging CPU architecture, holds vast potential in high-performance computing. However, current high-performance computing platforms are predominantly led by the X86 architecture. Commonly used models like WRF and CAMx have primarily been developed for X86-based platforms, lacking adaptation for emerging architectures. Therefore, the adaptation to specific RISC platform was approached comprehensively, considering architecture, compiler, compilation options, and model code. MIPS and LoongArch are both RISC architectures, and the Loongson 3A4000 CPU, representing MIPS processors, and the Loongson 3A6000 CPU, representing LoongArch processors, have been selected to enhance the scientific contribution of this manuscript. Through literature review, it was found that there is limited research on the application of scientific computing on MIPS and LoongArch platforms. Thus, our study has constructed a numerical modeling system with WRF-CAMx modelling system, which encompasses the runtime environment, models, control scripts, and environmental data. It can be practically utilized for operational numerical simulation, representing a relatively comprehensive scientific computing application. This study could provide methodological and technological references for scientific computing applications on MIPS and LoongArch platforms.

Referee 2: No new developments to either the WRF or CAMx models are described,

neither were any changes implemented to the respective model codes to improve their computational performance on the architectures examined. Rather changes were made to configuration/makefiles to facilitate the compiling of the model codes, which is somewhat standard practice whenever a model code is ported across platforms or when compilers are updated.

**Reply:** To build the WRF-CAMx modeling system on the MIPS platform, the developments in the WRF and CAMx model primarily include:

(1) Establish the runtime environment on emerging platforms, including parallel computing libraries such as MPICH3 (v3.4) and data format libraries such as HDF5 (v1.15.1) and NETCDF (C-v4.8.1, Fortran-v4.5.3). These libraries do not support the architecture (mips64el and LoongArch) and GNU compiler of Loongson platform. Relevant information needs to be added to the free software config.guess and config.sub provided by GNU org (part of the information is shown in Figure 3 for example, which can help identify the platform architecture and system during the compilation and installation of libraries using Configure and Make tools.

```
    ''
loongarch32:Linux:*:* | loongarch64:Linux:*:*)
    GUESS=$UNAME_MACHINE-unknown-linux-$LIBC
    ;;

    ''
mips64el:Linux:*:*)
    GUESS=$UNAME_MACHINE-unknown-linux-$LIBC
    ;;
```

Figure 3. Information about the architecture of Loongson platform

(2) Incorporate architecture-specific settings for the model. Taking the meteorological model WRF for example, the architecture presets are stored in the configure.defaults file, but seetings about the Loongson platform is not included. Specific architecture details, including GNU compiler and compilation options, need to be added, which can ensure the correct display of configuration during building WRF model, and part of information is shown in Figure 4 and Figure 5.

```
**ARCH    Linux mips64, gfortran compiler with gcc  #serial smpar dmpar dm+sm**
#
DESCRIPTION     =       GNU ($SFC/$SCC)
DMPARALLEL      =       # 1
OMPCPP          =       # -D_OPENMP
OMP             =       # -fopenmp
OMPCC           =       # -fopenmp
SFC             =       gfortran
SCC             =       gcc
CCOMP           =       gcc
DM_FC           =       mpif90 -f90=$(SFC)
DM_CC           =       mpicc -cc=$(SCC)
FC              =       CONFIGURE_FC
CC              =       CONFIGURE_CC
LD              =       $(FC)
RWORDSIZE       =       CONFIGURE_RWORDSIZE
PROMOTION       =       #-fdefault-real-8
ARCH_LOCAL      =       -DNONSTANDARD_SYSTEM_SUBR  -DWRF_USE_CLM
CFLAGS_LOCAL    =       -w -O3 -c
LDFLAGS_LOCAL   =
CPLUSPLUSLIB    =
ESMF_LDFLAG     =       $(CPLUSPLUSLIB)
FCOPTIM         =       -O2 -ftree-vectorize -funroll-loops
FCREDUCEDOPT    =       $(FCOPTIM)
FCNOOPT         =       -O0
```

Figure 4. Presets for the Loongson3A4000 platform.

```
* * *
Please select from among the following Linux mips64 options:

  1. (serial)   2. (smpar)   3. (dmpar)   4. (dm+sm)   GNU (gfortran/gcc)

Enter selection [1-4] : █
```

Figure 5. The display of WRF configuration on Loongson3A4000 platform

(3) Modify the code to make it run smoothly on a specific platform. Taking some function in the CAMx model for example, the model frequently uses the "write" function for formatted output. The format specifiers in the parameters consist of data types (I, F, E, A, X, etc.) followed by a character width. In the CAMx model, the format specifiers in the write function mostly default to character width, but there is a compilation issue with MIPS GNU, requiring character width descriptors. It is also essential to ensure consistency with the default precision. A specific example is illustrated in the figure below. A specific example is showed in Figure 6.

```
Before modification:
      write (iout,'(a,2a)') ' spec','total [ug/m3]','c* [ug/m3]    '
      write (iout,'(i5,2e)') (idx(i),sctot(i),scsat(i),i=1,nsol)
      write (iout,'(a,2e)') ' cpre,cpx ',cpre,cpx
After modification:
      write (iout,'(a5,2a15)') ' spec','total [ug/m3]','c* [ug/m3]    '
      write (iout,'(i5,2e15.7)') (idx(i),sctot(i),scsat(i),i=1,nsol)
      write (iout,'(a5,2e15.7)') ' cpre,cpx ',cpre,cpx
```

Figure 6. The modification of format specifiers in "write" function

However, the description of the modification and development of the model is not detailed enough in the manuscript, and some technical details have been added into Section 3 of the revised manuscript, which are as follows:

**lines 265 of original manuscript: 3 Porting the WRF-CAMx modelling system on MIPS CPU platform**

lines 306-307 of revised manuscript: *3 Porting the WRF-CAMx modelling system on MIPS and LoongArch CPU platforms*

[revised manuscript text omitted]

Furthermore, due to the hardware and ecosystem limitations of the MIPS platform, the improvement of model performance primarily relies on specific optimization options provided by the compiler based on the MIPS instruction set, which are mentioned in the compiler description of the manuscript. In the future, there will be considerations to modify the code for optimization.

Referee 2: It could be argued that running and porting of models across platforms and establishing the "reproducibility" of results through the benchmarking described falls under the scope of "development and technical papers", but there too the simulation durations and domain coverage are somewhat limited to clearly assess all technical aspects of running the models on the new architecture.

**Reply:** Due to the limited performance of the Loongson 3A4000 CPU at present, the simulation duration and domain coverage are somewhat restricted. Wang et al. (2019) used a 72h test case for validation of model results and analysis of computational performance. In this study, we set up a 72h simulation case covering the Beijing-Tianjin-Hebei region. The chosen computational scale and simulation duration are moderate for the platform, facilitating the comparative analysis and performance evaluation. This case serves as a representative example of short-term regional numerical simulation application, which is sufficient for validating the feasibility of numerical simulation on the MIPS platform. The cases of broader coverage and longer duration will be considered to test on platforms with more CPUs. Therefore, we have added the explanation in the revised manuscript, which are as follows:

**lines 162-164 of original manuscript:** For the meteorological model, the global meteorological initial and boundary fields for the WRF model are derived from the NCEP Global Final Reanalysis Data (FNL), with a spatial resolution of 0.5° x 0.5° and a temporal resolution of 6 hours.

lines 173-179 of revised manuscript: *In the research of Wang et al. in 2019, a 72h test case was set for the scientific validation and performance evaluation of the chemistry transport models. A 72h case represents a moderate-sized real scientific workload, which allows for simulating in a short time to validate the results and assess computational efficiency on the MIPS and LoongArch platforms. For the meteorological model, the global meteorological initial and boundary fields for the WRF model are derived from the NCEP Global Final Reanalysis Data (FNL), with a spatial resolution of 0.5° x 0.5° and a temporal resolution of 6 hours.*

lines 700-703 in the reference of revised manuscript: *Wang, H., Lin, J., Wu, Q., Chen, H., Tang, X., Wang, Z., Chen, X., Cheng, H., and Wang, L.: MP CBM-Z V1.0: design for a new Carbon Bond Mechanism Z (CBM-Z) gas-phase chemical mechanism architecture for next-generation processors, Geoscientific Model Development, 12, 749–764, https://doi.org/10.5194/gmd-12-749-2019, 2019.*

Referee 2: At several places in the manuscript discussion, it is mentioned that MIPS architectures and the Loongson 3A4000 have the advantage of energy efficiency. However, the simulation design (domain size and simulation duration) does not appear to lend itself to adequately assess potential energy savings. Neither is any analyses presented to robustly infer the tradeoff between computational performance (since that

seems to be poorer for the MIPS system used here relative to the X86 platform) and energy savings that may result from transitioning to such a platform.

**Reply:** In the manuscript, rough estimates have only been made through the CPU's TDP parameters and performance evaluation of specific case. However, compared to CISC architecture, the high energy efficiency of RISC is determined by the characteristics of instruction set. Therefore, MIPS has a natural energy efficiency advantage about the architecture compared to X86. In the future, we will try to find a MIPS platform with a similar number of CPU cores as the X86 platform to comprehensively evaluate potential energy savings. Currently, the MIPS platform lacks effective tools for energy efficiency assessment, thus, the statements in the abstract about the energy efficiency (lines 30-31 of original manuscript) has been deleted in the revised manuscript, which are as follows:

**lines 30-31 of original manuscript:** "*Thus, Loongson 3A4000 has higher 30 energy efficiency in the application of the WRF-CAMx modeling system*" **has been removed**.

L113-115: This statement implies that the WRF-CAMx modeling system was developed in Xi'an, China and Milan, Europe – is that an accurate representation of the origin of these models or their linkage? Did Ramboll not develop the requisite code to link CAMx with WRF output?

**Reply:** The WRF-CAMx modeling system mentioned in this manuscript refers to an operational system of numerical forecasting designed for specific region, not just a simple linkage of the WRF and CAMx models. There are also a series of tools which support the operation of the modeling system. Their functions mainly include: Firstly, the configuration of models, which primarily involves the unified settings for the simulation domains and grids for Beijing-Tianjin-Hebei region, and the parameterization schemes suitable for this region. Secondly, emission data processing. The tool of SMOKE is used to process the emission source data to the simulation grids. Additionally, the C Shell scripts are used for system control and modules interconnecting in the modeling system. For example, in our 'run_CAMx.csh' script, it will check whether the required data is prepared; if not, it returns to execute the pre-processing modules. The link program provided by Ramboll serves as a part of these tools. This statement may not be clear enough. Here, the references of the WRF-CAMx modeling system used in Xi'an, China, and Milan, Europe, indicate that our system presented in the manuscript holds significant application value. We have already revised the relevant statements in lines 113-115 of original manuscript according to your suggestions, and new statements are in lines 117-119 of revised manuscript, which are as follows:

**lines 113-115 of original manuscript:** *In Xi'an, China and Milan, Europe, the WRF-CAMx modelling system was developed, enabling high-resolution hourly model output*

*of pollutant concentration within specific local urban areas (Pepe et al., 2016; Yang et al., 2020).*

lines 117-119 of revised manuscript: *In Xi'an, China and Milan, Italy, the WRF-CAMx modelling system was applied, enabling high-resolution hourly model output of pollutant concentration within specific local urban areas (Pepe et al., 2016; Yang et al., 2020).*

L303: "stability and availability" should be clearly defined. Is a single 72-hour simulation duration sufficiently long to test the stability of a model on an architecture?

**Reply:** Since our platform is equipped with only one CPU, its computational capacity is limited. The simulation cases are configured with relatively short durations, providing a moderate computation scale for this platform. And a simulation duration of 72 hours is already sufficient to the short-term numerical forecasting, and also for testing the stability of the model system. This is recognized in the research and application of numerical models. In the research of Wang et al. in 2019, a 72h case was used for validation of model results and analysis of computational performance. We have added the explanation in the revised manuscript, which are as follows:

**lines 162-164 of original manuscript:** For the meteorological model, the global meteorological initial and boundary fields for the WRF model are derived from the NCEP Global Final Reanalysis Data (FNL), with a spatial resolution of 0.5° x 0.5° and a temporal resolution of 6 hours.

lines 173-179 of revised manuscript: *In the research of Wang et al. in 2019, a 72h test case was set for the scientific validation and performance evaluation of the chemistry transport models. A 72h case represents a moderate-sized real scientific workload, which allows for simulating in a short time to validate the results and assess computational efficiency on the MIPS and LoongArch platforms. For the meteorological model, the global meteorological initial and boundary fields for the WRF model are derived from the NCEP Global Final Reanalysis Data (FNL), with a spatial resolution of 0.5° x 0.5° and a temporal resolution of 6 hours.*

lines 700-703 of revised manuscript: *Wang, H., Lin, J., Wu, Q., Chen, H., Tang, X., Wang, Z., Chen, X., Cheng, H., and Wang, L.: MP CBM-Z V1.0: design for a new Carbon Bond Mechanism Z (CBM-Z) gas-phase chemical mechanism architecture for next-generation processors, Geoscientific Model Development, 12, 749–764, https://doi.org/10.5194/gmd-12-749-2019, 2019.*

L456-457: How does the parallel performance of CAMx vary with problem size, i.e., number of grid cells? What fraction of the time is spent in output operations? Is it possible that with increasing computational size, a single processor would require more

time than the configuration with two with one dedicated to I/O? How generalizable are the findings on parallel performance based on the limited domain size and simulation period?

**Reply:** In fact, the number of grid cells does objectively impact the model's parallelism, and the computational scale is one of the factors influencing the parallel performance of the model on a specific platform. However, due to our platform being equipped with one quad-core processor, and the limited computational scale of the test case, we have not encountered performance issues related to factors such as computational scale, I/O, and multiprocessors. Your considerations are comprehensive, and the questions are very meaningful, guiding us for further research in these directions. The statements about future research directions based on your guidance have been added into the revised manuscript, which are as follows:

**lines 490-492 of original manuscript:** The adaptation and optimization of the models based on MIPS CPUs will also be an important research direction in the future.

lines 551-555 of revised manuscript: *The adaptation and optimization of the models based on RISC CPUs will also be an important research direction in the future. Many factors influencing parallel performance, such as computing scale, I/O, multiprocessor, etc., will be considered to evaluate on platforms with stronger performance and more processors in the future.*

L468-474: It is interesting that the performance of the MIPS platform decreased significantly when the number of parallel processes exceed 4. Is this unique to the Loongson 3A4000 or is this generalizable to other MIPS systems? Would the same hold for a domain with significantly larger number of grid cells?

**Reply:** The reason for this phenomenon is that the Loongson 3A4000 CPU has only 4 cores, when running with four parallel processes, each core's usage approaches 100%. When the number of parallel processes exceeds 4, processes will compete for computing resources, resulting in additional overhead and performance degradation. If we are interested in this, we need more MIPS or LoongArch CPUs, but the prerequisite is to make the sponsor feel that it is a valuable thing. For other MIPS platforms, it depends on testing results. The computational performance of CAMx running with more than 4 parallel processes on Loongson 3A4000 and 3A6000 platforms have been added into the section 2 of the supplement and mentioned in the revised manuscript, which are as follows:

**Section 2 of supplement:** *Elapsed time of CAMx model on MIPS and LoongArch platforms*

[Figure]

***Figure S2.*** *Elapsed time of CAMx model running simulation case with MPICH for 24 hours on the MIPS and LoongArch platforms.*

**lines 529-531 of revised manuscript**: *This explains why the elapsed time is slightly higher when CAMx running with 5 parallel processes compared to 4 parallel processes as shown in the section 2 of Supplementary Material.*

**Response to Referee #3**

Thank you for the suggestions you provided. They are detailed and crucial for improving the manuscript.

**General comments**

Referee 3: The research paper focuses on the utilization of MIPS CPUs, particularly the Loongson 3A4000 CPU platform, in air quality prediction models. It evaluates the performance of the WRF-CAMx air quality modelling system in the Beijing-Tianjin-Hebei region using this platform. The study compares the MIPS CPU platform's performance with a benchmark X86 platform, analyzing various aspects like relative errors for major species, computational efficiency, and energy consumption. The results indicate the feasibility and efficiency of using MIPS architecture for such applications.

This work has the potential to offer valuable guidance for using the MIPS platform for geoscientific modeling. I would suggest that the authors provide a more in-depth discussion on how to exploit the advantages of the MIPS platform. The structure of the paper could be improved, and additional tests are necessary to demonstrate the MIPS platform's performance.

**Reply:** Thank you very much for your appreciation of the potential practical value of this research. The MIPS platform with Loongson 3A4000 CPU in the manuscript has limitations in hardware performance and software ecosystem. Firstly, the platform only supports the GNU compiler and specific Linux operating system currently. Therefore, performance evaluations can only be conducted in a certain environment. Secondly, due to the limited number of CPUs and cores on the platform, the maximum parallelism supported by the model system is 4. When the number of parallel processes exceeds 4, competition for computational resources restricts further performance evaluation of the platform. Additionally, the simulation case set up in this study fully utilize the computational resources of the platform, with the CPU utilization approaching 100% for each core during simulation. Therefore, performance evaluation with higher parallelism and larger computational scales is challenging on this platform. However, two parallel computing models supported by the platform were evaluated in the study: the process-based MPI model and the thread-based OpenMP model. This provides a certain level of representativeness, as illustrated in Figure 10 of the manuscript. What's more, the lack of effective softwares and tools to provide a more detailed assessment of the scientific computing capability on the platform is also an objectively existing issue. In the future, with the improvement of the ecosystem and performance of the MIPS platform, additional tests will be considered to further demonstrate the platform's performance and discuss how to exploit the advantages. Recently, we acquired a platform equipped with the Loongson 3A6000 CPU, which is the latest product released by Loongson Technology. In order to enhance the universality of our research, we conducted the same tests based on this platform. Relevant descriptions and results will be added to the revised manuscript. Additionally, the structure of the paper will be improved.

**Major comments and questions**

Referee 3: Every abbreviation that appears in the paper, including in the abstract and the main body, should be spelled out in full the first time it is used. For example, the abbreviations 'MIPS' and 'WRF-CAMx' are not spelled out in the abstract.

**Reply:** Thank you for the issues you pointed out. All abbreviations which appear for the first time will be carefully reviewed and corrected in the revised manuscript. We have spelled out the abbreviations in full in lines 14-16 and 18-22 of original manuscript, which are shown in lines 14-16 and 18-22 of revised manuscript as follows:

**lines 14-16 of original manuscript:** *The MIPS processor architecture is a type of Reduced Instruction Set Computing (RISC) processor architecture, which has advantages in terms of energy consumption and efficiency.*

lines 14-16 of revised manuscript: *The Microprocessor without interlocked piped stages (MIPS) and LoongArch are Reduced Instruction Set Computing (RISC) processor*

*architectures, which have advantages in terms of energy consumption and efficiency.*

**lines 17-19 of original manuscript:** *In this study, Loongson 3A4000 CPU platform with MIPS64 architecture was used to establish the runtime environment for the air quality modelling system WRF-CAMx in Beijing-Tianjin-Hebei region.*

lines 18-22 of revised manuscript: *In this study, Loongson 3A4000 CPU platform with MIPS64 architecture and Loongson 3A6000 CPU platform with LoongArch architecture were used to establish the runtime environment for the air quality modelling system Weather Research and Forecasting–Comprehensive Air Quality Model with extensions (WRF-CAMx) in Beijing-Tianjin-Hebei region.*

Referee 3: The model setups and analysis methods used in this paper should be presented prior to the results in Section 4. The content in Lines 305-309, 323-325, and 404-411 should be consolidated in Sections 2 or 3 as part of the methodology.

**Reply:** Thank you for the suggestions about the organization and structure of the article. These will be carefully considered and reflected in the improvement of the revised manuscript. We have moved the contents in lines 304-309 of original manuscript to lines 171-177 in section 2.1.2 of revised manuscript, lines 323-325 of original manuscript to lines 134-136 in section 2.1 of revised manuscript, lines 398-411 of original manuscript to lines 194-208 in the section 2.1.3 of revised manuscript.

**lines 304-309 of original manuscript:** *Starting from 00:00 on November 3, 2020, until 24:00 on November 5, 2020, the modelling system simulated the meteorological and air quality for a period of 72 hours, represents a moderate-sized real scientific workload, which allows for testing in a short time, and validating the results of the WRF-CAMx model on the MIPS platform and assessing computational efficiency.*

lines 171-177 of revised manuscript: *Starting from 00:00 on November 3, 2020, until 24:00 on November 5, 2020, the modelling system simulated the meteorological and air quality for a period of 72 hours. In the research of Wang et al. in 2019, a 72h test case was set for the scientific validation and performance evaluation of the chemistry transport models. A 72h case represents a moderate-sized real scientific workload, which allows for simulating in a short time to validate the results and assess computational efficiency on the MIPS and LoongArch platforms.*

**lines 134-136 of revised manuscript:** *The relative humidity, a meteorological variable used in result validation is calculated using the wrf-python package (Official website: https://wrf-python.readthedocs.io, last access: October 2023).*

**lines 194-208 of revised manuscript:**

**2.1.3 Statistical indicators for model results**

*To quantify the differences in the model results between the MIPS and benchmark platform, three statistical indicators are used to analyze the differences of concentration time series: Mean Absolute Error (MAE), Root Mean Square Error (RMSE), and Mean Absolute Percentage Error (MAPE). The MAPE quantifies the deviation between computational differences and simulated values. The smaller these indicators, the better accuracy and stability of scientific computing of the modeling system on the MIPS platform. The calculation formulas for these statistical indicators are provided in equations (1) to (3).*

$$MAE = \frac{1}{n}\sum_{i=1}^{n}|MIPS(i) - Base(i)| \tag{1}$$

$$RMSE = \left[\frac{1}{n}\sum_{i=1}^{n}(MIPS(i) - Base(i))^2\right]^{\frac{1}{2}} \tag{2}$$

$$MAPE = \frac{1}{n}\sum_{i=1}^{n}\left|\frac{MIPS(i) - Base(i)}{MIPS(i)}\right| \times 100\% \tag{3}$$

*In the equations, n represents the number of grids in the domain. MIPS(i) represents the simulated value of a certain grid on the MIPS platform, and Base(i) represents the baseline value of a certain grid on the benchmark platform.*

Referee 3: I am curious about the number of sockets available on the motherboard for the Loongson 3A4000 platform. Could the author conduct a larger-scale comparison using more Loongson 3A4000 CPUs compared to the X86 platform as shown in Figure 9?

**Reply:** The motherboard of the Loongson 3A4000 platform used in this study supports only one CPU with four physical cores. We will consider seeking platforms with more Loongson 3A4000 CPUs for larger-scale tests.

Referee 3: Could the author investigate the impact of using different compilers or different compiler parameters on computational performance, in addition to the GNU?

**Reply:** Currently, the MIPS platform with Loongson 3A4000 CPU only supports the GNU compiler. Other compilers have not been adapted for the MIPS architecture. As for compiler parameters, the default options for MIPS GNU are fixed. They are used to specify the architecture of the target platform and optimize the target program based on specific instruction sets. There are no compiler parameters that significantly impact computational performance.

**Minor comments**

Referee 3: Line 92: Remove "The remainder is organized as follows."

**Reply:** This statement has been removed in the revised manuscript.

Referee 3: Line 113-115: Rephrase this sentence. The WRF is developed by NCAR, and CAMx is developed by Rambell. WRF-CAMx is just applied in Xi'an, China and Milan, Italy (not Europe).

**Reply:** This part of the statement is not clear and accurate enough. The intended meaning here is that the WRF-CAMx modeling system suitable for specific regions (such as Xi'an and Milan) has been applied in the research cited in the references. We have already revised the relevant statements in lines 113-115 of original manuscript according to your suggestions, and new statements are in lines 117-119 of revised manuscript, which are as follows:

**lines 113-115 of original manuscript:** *In Xi'an, China and Milan, Europe, the WRF-CAMx modelling system was developed, enabling high-resolution hourly model output of pollutant concentration within specific local urban areas (Pepe et al., 2016; Yang et al., 2020).*

lines 117-119 of revised manuscript: *In Xi'an, China and Milan, Italy, the WRF-CAMx modelling system was applied, enabling high-resolution hourly model output of pollutant concentration within specific local urban areas (Pepe et al., 2016; Yang et al., 2020).*

Referee 3: Line 123-126: The introduction for WRF is not professional. WRF is a meso-scale meteorology model, and you can use it for weather research and prediction. It can be used with a data assimilation technique, and testing its parameterization schemes is a way to improve WRF.

**Reply:** The description about the WRF model is not detailed and accurate enough, and it will be improved based on your suggestions in the revised manuscript. The introduction for WRF has been revised in lines 123-126 in section 2.1 of original manuscript, which are shown in lines 127-132 of revised manuscript as follows:

**lines 123-126 of original manuscript:** *WRF is a high-resolution mesoscale model, which can be utilized for various purposes such as weather research and forecasting, physical parameterization scheme research, data assimilation and mesoscale climate simulation.*

lines 127-132 of revised manuscript: *WRF is a mesoscale numerical weather prediction system designed for atmospheric research and operational forecasting applications.*

*Distinguished by its high temporal and spatial resolution, WRF is suitable for multi-scale simulations of short-term weather forecast, atmospheric process, and long-term climate, making it an essential tool in the meteorological and atmospheric research communities (Powers et al., 2017).*

**Referee 3: Line 152-154: Why you used 14 layers not original 34 layers?**

**Reply:** In practical applications of air quality simulation, we primarily focus on the concentration of air pollution species near the land surface. Therefore, the analysis of simulation results often involves extracting data from the near-surface layer for evaluation. There is no need for an excessive number of vertical layers in the simulation cases of the CAMx model. This approach can significantly save computational resources, particularly in situations where resources on MIPS platforms are limited. Merging the vertical layers appropriately is more conducive to tests.

**Referee 3: Line 195-196: Was FFT used or related to this paper? If not, please remove it.**

**Reply:** FFT was not used in this paper. The citation is intended to illustrate that there has been previous research on the application of scientific computing on MIPS platforms with Loongson CPUs. However, there is scarce research specifically focusing on large-scale applications like numerical models, with most studies centered around specific algorithms or programs (such as FFT). Therefore, we can only reference these studies. This part has been removed in the revised manuscript.

**line 193-200 of original manuscript:** "A lot of porting and optimization research work has been conducted to ensure the proper functioning of the high-performance mathematical library on Loongson platforms, resulting in improved computing performance, such as FFT (Fast Fourier Transform) (Guo et al., 2012; Li et al., 2011; Zhao et al., 2012). The porting and optimization efforts conducted on the multi-core Loongson processors have successfully demonstrated the stability and efficiency in the numerical computing applications. These results provide valuable technical references and rationality validation for the numerical model application on Loongson platform." has been removed. And the references "Guo et al., 2012", "Li et al., 2011", and "Zhao et al., 2012" has been deleted in the revised manuscript.

**Referee 3: Line 436: Give full names to RMSE, std. Why std are in lowercase but RMSE is not? Also, what's the statistic meaning or purpose of using the ratio of RMSE/STD?**

**Reply:** The full names of RMSE (Root Mean Square Error) and std (standard deviation) will be added to the revised manuscript. The lowercase of std does not have a specific meaning. It is only used to distinguish it from statistical indicators such as MAE and

RMSE, which are applied to analyze simulation differences between platforms. 'std' is only used to describe the dispersion of a certain species' concentration in the simulation results. The ratio of RMSE/ STD does not have a specific statistical meaning. Wang et al. (2021) introduced it as a metric to assess the scientific usability of simulation results after model improvement. This has been recognized in academic research. In our study, the extremely small ratio of RMSE/STD means the simulation differences between the two platforms are negligible compared to the spatial variations of simulation results. The full names of RMSE (Root Mean Square Error) and std (standard deviation) have been added in lines 428-431 of original manuscript, which are shown in lines 474-478 of revised manuscript as follows:

**lines 428-431 of original manuscript:** *The RMSEs for NO2, SO2, O3, CO, PNO3 and PSO4 concentration between the MIPS and benchmark platform were computed, along with the standard deviation used to describe the spatial variation of species, and the ratio of RMSE to std, as shown in Table 6.*

lines 474-478 of revised manuscript: *The Root Mean Square Errors (RMSEs) for $NO_2$, $SO_2$, O3, CO, $PNO_3$ and $PSO_4$ concentration between the MIPS and benchmark platform were computed, along with the standard deviations (stds) used to describe the spatial variation of species, and the ratio of RMSE to std, as shown in Table 6.*

**Refernces:**

Wang, H., Lin, J., Wu, Q., Chen, H., Tang, X., Wang, Z., Chen, X., Cheng, H., and Wang, L.: MP CBM-Z V1.0: design for a new Carbon Bond Mechanism Z (CBM-Z) gas-phase chemical mechanism architecture for next-generation processors, Geoscientific Model Development, 12, 749–764, https://doi.org/10.5194/gmd-12-749-2019, 2019.